

# On the regional-scale streamflow variability using flow duration curve

Pankaj Dey[1], Jeenu Mathai[2], Murugesu Sivapalan[3,4] and Pradeep. P. Mujumdar[5,6]

[1]Department of Hydrology, Indian Institute of Technology, Roorkee, India

[2]National Centre for Earth Science Studies, Thiruvananthapuram, India

[3]Department of Civil and Environmental Engineering, University of Illinois at Urbana-Champaign, Urbana, IL, USA

[4]Department of Geography and Geographic Information Science, University of Illinois at Urbana-Champaign, Urbana, IL, USA

[5]Department of Civil Engineering, Indian Institute of Science, Bangalore, India

[6]Interdisciplinary Centre for Water Research, Indian Institute of Science, Bangalore, India

*Correspondence to*:  Pankaj Dey (pdey609@gmail.com)

**Abstract.** As each catchment responds uniquely, even if they appear similar, formulating generalizable hypotheses and using routinely used signatures of catchment similarity to examine streamflow variability can be difficult. Flow Duration Curve (FDC), a concise portrayal of streamflow variability at a specific gauging station, can provide insights into hydroclimatic and landscape processes occurring at a wide range of space and time scales that govern flow regimes in a region. This study explores the suitability of partitioning of annual streamflow FDC into seasonal FDCs, and total streamflow FDC into fast and slow flow FDCs to unravel the process controls on FDCs at a regional scale, with application to low-gradient rivers flowing east from the Western Ghats of Peninsular India. The focus is on investigation of the controls of common regional landscape features (in space) and seasonal climatic (in time) variations on regional variations of the FDC. Findings of the study indicate that bimodal rainfall seasonality and higher fraction of moderate to good groundwater potential zones explains the higher contribution of slow flow to total flow across north-south gradient of the region. Shapes of fast and slow FDCs are controlled by recession parameters revealing the role of climate seasonality and geologic profiles, respectively. A systematic spatial variation across north-south gradient is observed– highlighting the importance of coherent functioning of landscape-hydroclimate settings in imparting distinct signature of streamflow variability. The framework is useful to discover the role of time and process controls on streamflow variability in a region with seasonal hydroclimatology and hydro-geologic gradients.

## 1 Introduction

The hydrologic functioning of catchment systems in any given region is coevolved with the long-term climatology and landscape features present in the region through mutual interactions operating across multiple spatial and temporal scales (Wagener et al., 2013). These interactions and long-term feedbacks impart variability to hydrologic processes that are characteristic of the region of interest, including runoff generation and riverine transport processes, thus influencing water availability and reliability to human populations that depend on the streamflow. Understanding streamflow variability in time and space across river basins in the region is therefore





very important for water resource management (Deshpande et al., 2016; Sinha et al., 2018) and the prediction and
mitigation of floods (Kale et al., 1997). The frequency of high flows, low flows, or flows within specific ranges,
is essential for risk assessment of water management projects involving control of streamflow variability. Correct
portrayal of streamflow variability at the scale of catchments and river basins is therefore an indispensable
component in many hydrologic applications.
The focus of this paper is on the flow duration curve (FDC), which is a compact description of temporal
streamflow variability at the catchment scale. The FDC represents (daily) streamflow values plotted against the
proportion of time the given flow is exceeded or equalled (Smakhtin, 2001; Vogel & Fennessey, 1994). The
graphical form of the FDC embeds within it the governing hydrologic processes and dominant flow characteristics
throughout the range of recorded streamflows at the catchment scale (Botter et al., 2008). In this sense, the FDC
is also an important signature of a catchment's rainfall to runoff transformation (Ghotbi et al., 2020a; Vogel &
Fennessey, 1994). FDC thus typifies the old proverb, "one picture is worth a thousand words" with its potential
to encapsulate much of the relevant information of streamflow variability in a single plot (Vogel & Fennessey,
1995), and has been used in many hydrologic applications. Vogel and Fennessey (1994) provide a brief history of
the application of flow duration curves in hydrology. Applications of FDC include waste load allocation (Searcy,
1959), water quality management (Searcy, 1959; Rehana & Mujumdar, 2011, 2012), reservoir and sedimentation
studies (Vogel & Fennessey, 1995), low-flow and flood analyses (Smakhtin, 2001), assessment of environmental
flow requirements (Smakhtin and Anputhas, 2006), and water availability for hydropower (Basso & Botter, 2012).
Streamflow observed in a river is the culmination of interacting hydrological processes of runoff generation,
overland and subsurface flow and evaporation, operating at multiple time and space scales, in response to climatic
inputs and their interactions with a range of landscape properties, all of which are highly heterogeneous. This
makes it challenging to decipher the process controls on streamflow variability, and their manifestation in the
shape of the FDC (Cheng et al., 2012; Ghotbi et al., 2020b; Yokoo & Sivapalan, 2011). Therefore, there is a need
for appropriate conceptual frameworks that can bring out these process controls of FDCs and generate deep
insights into the governing principles underpinning observed variability. Yokoo and Sivapalan (2011) presented
a framework for deciphering the process controls of the FDC by considering the FDC of total streamflow (TFDC)
as a statistical summation of a fast flow duration curve (FFDC) and a slow flow duration curve (SFDC). FFDC is
a filtered version of precipitation variability, with rainfall intensity patterns and surface soil characteristics as
controlling factors (Yokoo & Sivapalan, 2011). On the other hand, SFDC reflects a competition between
subsurface drainage and evapotranspiration (Yokoo & Sivapalan, 2011), in which case seasonality and regional
geology are stronger controlling factors. This contrast in the process controls governing quick (surface) runoff
and slow (subsurface) flow, supports the notion of stratifying total streamflow into these two components
operating at two different time scales. The distinction between the two (fast and slow) flow time scales enables
the conceptualization of the process controls of fast flow (surface runoff) and slow flow (subsurface streamflow
and groundwater flow) separately (Cheng et al., 2012; Yokoo & Sivapalan, 2011).
Ghotbi et al (2020a, 2020b) used this framework to explore the climatic and landscape controls of FDCs using
streamflow data for hundreds of catchments across the continental United States in a comparative manner.  In
their work Ghotbi et al. (2020a) emphasized the need to consider the fast flow and slow flow time series
independently as stochastic responses of catchments to sequences of storm events. Intensity and frequency of





rainfall events and the properties of soils and topography govern the variability of fast flows, whereas climate seasonality and regional geology of the aquifer system govern variability of slow flow components. More specifically, Ghotbi et al. (2020b) showed the dominant process controls of FDCs as aridity index, topographic slope, coefficient of variation of daily precipitation, timing of rainfall, time interval between storms, snow fraction, and recession slope.

Due to significant differences between fast and slow process controls, each may be used to explain streamflow variability independently. While recognizing the necessity to represent the hydrological processes across two distinct time scales, this paper aims to develop a process-based understanding of how regional scale features impact streamflow variability across Peninsular India, using the flow duration curve (FDC) as a signature of this variability. For this purpose, an extension of this concept was made by including seasonal (timing) streamflow variability in a regional context. To isolate the effects of the drivers on the observed FDCs and to identify the controls of time and process scales on streamflow variability, a modeling framework is presented that comprises partitioning streamflow in multiple ways: seasons/months in the time domain, east-west/north-south directions in the space domain, fast/slow flows in the process domain. Streamflow data available from a large number of stream gauges within and between the major river basins across Peninsular India is employed for this purpose. The scientific novelty and methodological advancement of the paper lie in two interconnected aspects, which have not been adopted in the literature to date: (i) the timescale partitioning framework is used to study the relative contributions of different seasons to the FDC (repeated for fast and slow flow components), exploring how the relative contributions holistically vary across the whole region and using the framework to reconstruct the annual flow duration curve using seasonal flow duration curves, (ii) the Wegenerian approach in connecting the spatial variability of streamflow at a regional scale using flow duration curve. Thus, the main goal of this paper is to reconstruct the flow duration curves at different scales to unravel the regional scale streamflow variability by extending the process partitioning (Ghotbi et al., 2020a) with the time partitioning. Studies that use simultaneous partitioning of flow duration curves at seasonal and process scales to investigate regional streamflow variability in space and seasonal climatic in time fluctuations using the Wegenerian approach are limited. The remainder of the paper is structured as follows. Section 2 elaborates on the details of the study area and the daily streamflow dataset used. The description of the conceptual framework employed for the analysis is presented in Section 3. The results of the application of the framework to Peninsular India and the interpretation of the results are presented in Sections 4 and 5, respectively. Finally, the paper is concluded in Section 6 with key insights gained for the nature and controls of streamflow variability across Peninsular India.

**2 Study region**

Peninsular India is a cratonic region with an approximate shape of a vast inverted triangle with diverse topography and characteristic climatic patterns, bounded by the Arabian Sea in the west, the Bay of Bengal in the east, and the Vindhya and Satpura ranges in the north. The long escarpments of the Western Ghats and the Eastern Ghats, constituting the western and eastern continental fringes of India, and an asymmetric relief with eastward tilt towards the floodplains of several eastward draining rivers from the 1.5 km high Western Ghats, characterize the physiography of Peninsular India (Richards et al., 2016).





The rise of the Himalayan-Tibetan plateau has significantly contributed to the Neogene climate of Asia, favoured
the birth of the modern monsoon (Fig. 1.a, b) (Chatterjee et al., 2013, 2017), and triggered glaciation in the
Northern region. A wide variety of plateaux, open valleys, bedrock gorges, mountain ranges, inselbergs and
residual hills constitute the geomorphology of Peninsular India (Kale & Vaidyanadhan, 2014). The Peninsular
landscape is dominated by Deccan Traps (Deccan basalts) of Cretaceous-Eocene, igneous and metamorphic rocks
(Granite-gneisses) of Archaean-Late Precambrian along with minor consolidated sediments (Sandstone, shale) of
Precambrian-Jurassic (Fig. 1.c) (Kale, 2014).
The region is strongly impacted by monsoons, major seasonal winds which are a manifestation of the seasonal
movement of the Intertropical Convergence Zone (ICTZ in Fig. 1.a and Fig. 1.b), which contribute largely to the
annual rainfall variability in much of the Indian subcontinent (Gadgil, 2003). The monsoons have two components
– South-West monsoon and North-East monsoon, which arrive during June – September (JJAS) and October –
December (OND), respectively. South-West monsoon season contributes more than 75% of annual rainfall over
majority of the regions of the country (Saha et al., 1979). However, the Southern Peninsula receives a significant
portion (30-60%) of its annual rainfall during the North-East monsoon, which contributes only 11% of the rainfall
annually to India as a whole (Rajeevan et al., 2012). The maximum extent of rainfall over the Southern Peninsula
during the North-East Monsoon is due to the reversal of lower-level winds over South Asia from the South-West
to the North-East during the retreating phase of the South-West monsoon (Rajeevan et al., 2012). In Peninsular
India, there is a spatial variability of the South-West monsoon in the south-north direction. For example, the
Western Ghats, located at the western edges of Krishna and Cauvery basins, obstruct the incoming South-West
monsoon winds causing heavy rainfall on the mountains. After crossing the Western Ghats, the monsoon winds
have less moisture, causing a sharp decline in rainfall amounts towards the central and the north-eastern part of
the Peninsula (Fig S2.a in Supplementary Material). The North-East monsoon occurs during winter, and mostly
influence the rainfall in the Cauvery and some parts of the Krishna basins. Vegetation on the long escarpment of
Western Ghats is primarily tropical evergreen forest, which plays an important role in intercepting the South-West
monsoon winds (Ramachandra, 2018). Ramachandra (2018) portrayed the profile of vegetation across the west-
east gradient as it varies from tropical-evergreen to semi-evergreen and then moist to dry deciduous forests
towards the rain-shadow region just east of the Western Ghats. The topography map for the Peninsular region and
a selected point in the region is depicted in Fig. S1.a and Fig. S1.b in Supplementary Material, respectively. The
western margin of Peninsular India experiences heavy rainfall due to the presence of Western Ghats, whereas the
rain shadow region witnesses deficient rainfall (Fig. S2.c). It can thus be seen that the long geological, tectonic
history and the onset of monsoon climate events have made an imprint in the shaping the present landform of the
Indian Peninsula (Kale, 2014).

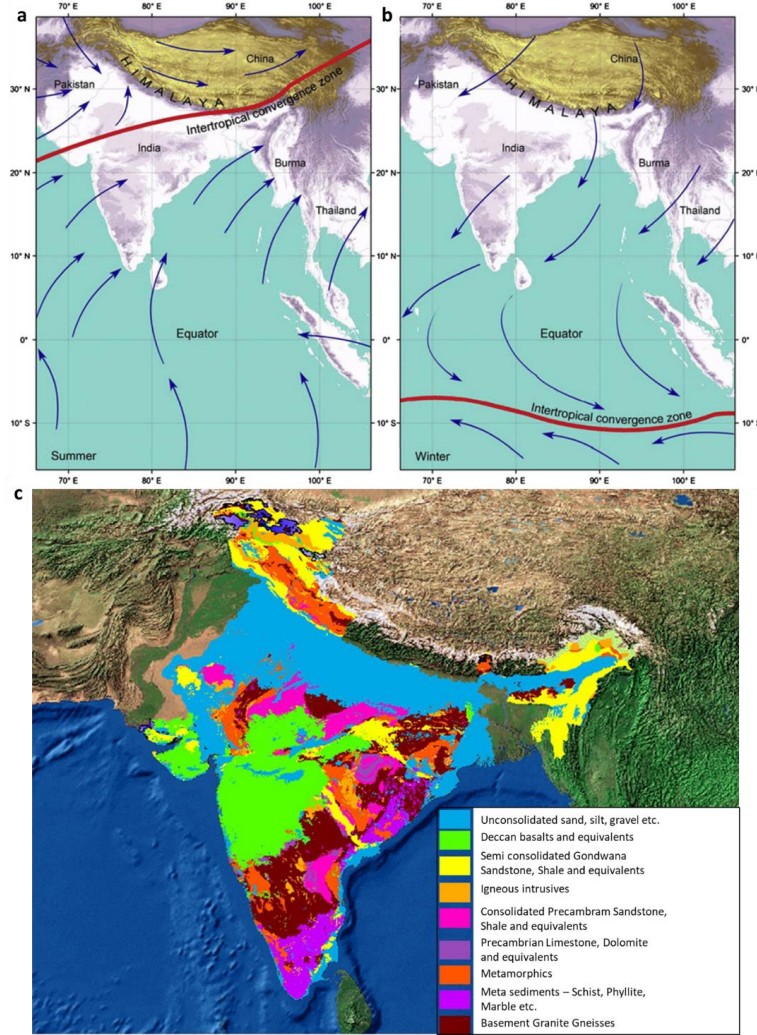



**Figure 1.** (a) The relation of uplift of Himalaya-Tibetan Plateau and monsoon initiation in India. Monsoon winds
blow from the Indian Ocean towards land in the summer (b) during the winter, the Himalaya prevents cold air
from passing into the subcontinent and causes the reversal of wind direction and monsoon blow from land toward
sea [Reprinted from (Chatterjee et al., 2013)] (c) geology of Peninsular India [Reprinted from: Central Ground
Water Board(https://www.aims-cgwb.org/general-background.php)].


The region shown in Fig. 2 is selected as the study area in the Deccan Plateau of Peninsular India. The escarpment
of Western Ghats forms the western margin of the Deccan Plateau which serves as the main water divide for the
Peninsular River systems. The gentle slope from west to east causes Peninsular rivers such as the Mahanadi,
Godavari, Krishna, and Cauvery (Fig. 2) to flow eastwards. Three of these rivers (Godavari, Krishna and Cauvery)
originate from the Western Ghats, spread across the area from the Deccan Plateau, flow eastwards, and drain into
the Bay of Bengal. The Mahanadi River rises in the mountains of Siwaha bounded by the Eastern Ghats in the
south and east, and drain eastwards into the Bay of Bengal. The Mahanadi basin constitutes a total catchment area
of about 141,600 km$^2$ with an average annual rainfall of 1,360 mm and a mean annual river flow of 66,640 million
m$^3$ (Rao et al., 2017). With an annual average rainfall of 1096 mm, the Godavari, the largest of all Peninsular
rivers, receives nearly 84 percent of its annual rainfall on average during the South-West monsoon (Koneti et al.,
2018). The Godavari basin's challenges include frequent flooding in its deltaic lower reaches, given the area's
proximity to the coastal zone, which is prone to cyclones, and frequent drying up during the drier months (Koneti
et al., 2018). Krishna is Peninsular India's second-largest river, with a total catchment area of 2,60,000 km$^2$, and
is susceptible to floods and droughts in some specific regions (Chanapathi & Thatikonda, 2020). The South-West
monsoon is the most significant contributor to rainfall in the Krishna basin, accounting for about 90% of its total
rainfall; the Krishna Basin, however, has a non-uniform rainfall distribution caused by climate variability, with
an average annual rainfall bout of 770 mm (Chanapathi & Thatikonda, 2020). Annual rainfall in the Cauvery
varies from 621 mm in the lower reaches to 4137 mm in the mountainous uplands, exhibiting considerable
variation across the basin (Kumar Raju & Nandagiri, 2017). The river Krishna, with a mean annual runoff of less
than 100 mm, is designated as an arid river (Milliman JD, 2011; Gupta et al., 2022), Cauvery as a semiarid river
(100-250 mm), Mahanadi and Godavari as humid rivers (250–750 mm). The higher baseflow index occurs within
0.5 and 0.7 in catchments in the Godavari and Mahanadi basins, whereas the lower baseflow index is noted from
0.25 and 0.45 in the Cauvery and Krishna basins(Bhardwaj et al., 2020). For agricultural purposes, the semiarid
regions of the Cauvery basin rely more on groundwater than surface water when compared to the other three
basins (Sreelash et al., 2020).
In this study, daily streamflow data between 1965 to 2012 for 62 stream gauges (Fig. 2) are selected from Water
Resources Information System database (WRIS) and located across the four river basins. The daily gridded rainfall
product at spatial resolution of 0.25° × 0.25° from India Meteorological Department (IMD) is also employed for
the analysis (Pai et al., 2014).



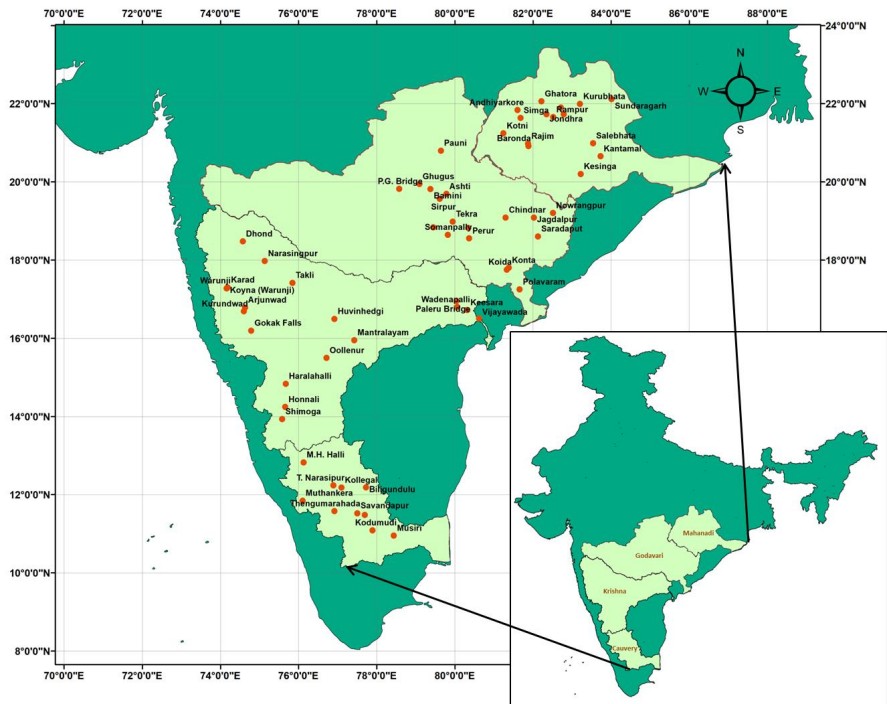

**Figure 2.** Location map of four Peninsular River Basins. Stream gauges considered in this study are marked with red circles.

### 3 Conceptual framework for stratification of streamflow variability using time scale

In this section, we check the suitability of a framework to stratify observed streamflow time series in the time domain into distinct time scales to better understand the physical controls of streamflow variability across the region. Partitioning of streamflow across seasonal and monthly time scales is able to bring out the role of climate seasonality on streamflow variability. Moreover, the progression of the seasons spatially imparts signatures on streamflow variability regionally as a whole. Time scale partitioning thus offers an opportunity to understand these climatic and landscape controls on streamflow variability through quantifying the relative contributions of seasonal streamflow on annual streamflow variability and how they vary regionally.

The streamflow hydrograph is the response of a physical, deterministic system (catchment) to a sequence of rainfall events. Given that the rainfall events are very much random in all their properties, equivalently, the streamflow hydrograph can also be seen as a stochastic time series, with streamflow considered a random variable. Therefore, it is amenable to a stochastic treatment in terms of distribution functions (e.g., cumulative distribution function, CDF). A major advantage of the CDF is that it enables us to make a concise statement of streamflow variability across a population of events. They have diagnostic value in that they can explain or interpret a catchment's streamflow response and compare it across many catchments and they help to classify catchments based on the flow regimes. They also have practical value in engineering design and environmental monitoring





that require a probabilistic treatment of streamflow. The cumulative distribution function of a random variable
(the random variable of interest to us is daily streamflow; $Q$) expresses the probability that a realization (i.e.,
observation) of $Q$ does not exceed a specific value $q$:
Cumulative Distribution Function (CDF):

$$F(q) = P[Q \leq q] \tag{1}$$

The flow duration curve is an alternative, but equivalent, measure of the streamflow variability that is widely used
in hydrology. The flow duration curve is a plot that shows the fraction of time ($D$) that the streamflow is likely to
equal or exceed some specified value of interest. Mathematically, $D$ can be expressed as,

$$D(q) = P[Q \geq q] = 1 - F(q) \tag{2}$$

Despite its probabilistic definition given above, in hydrological applications, the flow duration curve is plotted in
terms of $q(D)$ i.e., $q$ (in the vertical axis) as a function of $D$ (in the horizontal axis).
*Time scale partitioning of streamflow variability*
The streamflow time series can be equivalently divided into temporal segments of distinct seasons as well as
distinct months. In this case, by joining observed time series over multiple years, FDCs for each time segment can
be reconstructed. Assuming independence (as an approximation), these can then be combined to generate annual
FDCs. The theory for the time scale partitioning is illustrated in Fig. 3. The year is divided into three distinct (non-
overlapping) seasons, viz. Non-monsoon, South-West, and North-East seasons (for Peninsular India) of relative
durations $\tau_1$, $\tau_2$, and $\tau_3$ (with $\tau_1 + \tau_2 + \tau_3 = 1$) respectively. These seasons can be assumed to have distinct
characteristics in terms of rainfall variability and how they translate to streamflow variability. The daily
streamflow time series is used to construct the seasonal as well as annual FDCs. For example, the FDC of Non-
monsoon season is constructed by using the daily streamflow during the period of January – May over the years.
Similarly, FDCs for South-West and North-East monsoons are constructed using the daily streamflow during June
– September and October – December months over the years respectively and the annual FDC is constructed using
daily streamflow values for 365/366 days over the years. The FDCs at monthly time scales are obtained using the
daily values of streamflow in a month over the years. The FDCs for the three distinct seasons, i.e., Non-monsoon,
South-West monsoon, North-East monsoon, are denoted as $D_{NM}(q)$, $D_{SW}(q)$, and $D_{NE}(q)$ respectively. Initially,
the FDCs for each season can be constructed separately (Fig. 3).





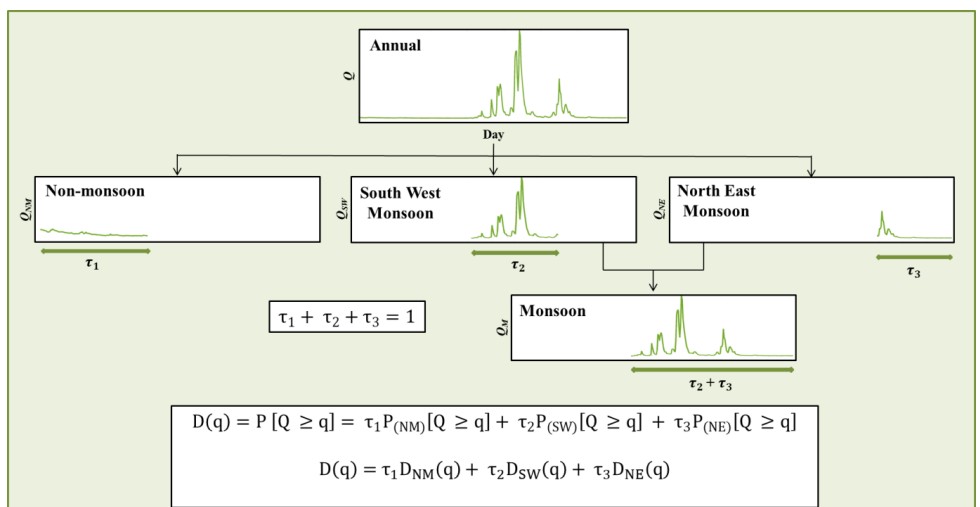


**Figure 3.** Scale partitioning into seasonal and monthly time scales. The conceptual framework illustrates the time
scale partitioning of streamflow time series into various seasonal components considering patterns of rainfall
variability. The annual streamflow time series is decomposed into three components: (1) Non-monsoon flow, (2)
South-West monsoon flow, and (3) North-East monsoon flow.


The annual FDC with exceedance probability $P\,[Q \geq q]$ refers to the probability of flow in annual scale being
greater than or equal to q, and is expressed as

$$D(q) = P\,[Q \geq q] = \tau_1 P_{(NM)}[Q \geq q] + \tau_2 P_{(SW)}[Q \geq q] + \tau_3 P_{(NE)}[Q \geq q] \qquad (3)$$

$$\text{or,} \quad D(q) = \tau_1 D_{NM}(q) + \tau_2 D_{SW}(q) + \tau_3 D_{NE}(q) \qquad (4)$$

where, $P_{(NM)}[Q \geq q]$, $P_{(SW)}[Q \geq q]$ and $P_{(NE)}[Q \geq q]$ refer to, respectively, the probability of flow in Non-
monsoon, South-West monsoon and North-East monsoon being greater than q. As the seasons are non-
overlapping, the probability of flow being greater than q at annual scale (i.e., $P\,[Q \geq q]$) can be expressed as the
sum of the weighted probabilities of flow being greater than q in the three seasons.
In general, the FDC at the annual scale can be constructed as follows:

$$D(q) = \tau_1 D_1(q) + \tau_2 D_2(q) + \cdots + \tau_n D_n(q) \qquad (5)$$

where $n$ is the number of distinct seasons considered for the analysis and, $\tau_1 + \tau_2 + \cdots + \tau_n = 1$. The validity of
the above depends on the assumption that there is no carryover of flows from one season to the next season (which





is an approximation). In this study, the assumption of independence between flows across three seasons is checked
using multivariate Hoeffding's test (see details in Text S1 of Supplementary Information).
If $F_A(.), F_{NM}(.), F_{SW}(.)$ and $F_{NE}(.)$ represent cumulative distribution function of daily flows during annual, Non-
monsoon, South-West monsoon and North-East monsoon, respectively, then using equation (2), equation (6) can
be written as:

$$1 - F_A(q) = \tau_1[1 - F_{NM}(q)] + \tau_2[1 - F_{SW}(q)] + \tau_3[1 - F_{NE}(q)] \tag{6}$$

Differentiating the above equation with respect to $q$ ,

$$f_A(q) = \tau_1 f_{NM}(q) + \tau_2 f_{SW}(q) + \tau_3 f_{NE}(q) \tag{7}$$

where $f_A(.), f_{NM}(.), f_{SW}(.)$ and $f_{NE}(.)$ represent probability density functions of annual, Non-monsoon, South-
West monsoon and North-East monsoon flows respectively.
If $Q, Q_{NM}, Q_{SW}$ and $Q_{NE}$ represent random variables comprising of daily streamflow at annual, Non-monsoon,
South-West monsoon and North-East monsoon time scales respectively, the expectation $E(Q)$ and variance $V(Q)$
of annual flow in terms of seasonal flows can be expressed as

$$E(Q) = \tau_1 E(Q_{NM}) + \tau_2 E(Q_{SW}) + \tau_3 E(Q_{NE}) \tag{8}$$

$$V(Q) = \tau_1 E(Q_{NM}^2) + \tau_2 E(Q_{SW}^2) + \tau_3 E(Q_{NE}^2) - \left(E(Q)\right)^2 \tag{9}$$

The magnitudes of $\tau_1$, $\tau_2$ and $\tau_3$ are $\frac{5}{12}, \frac{4}{12}$ and $\frac{3}{12}$ based on the annual proportions of Non-monsoon, South-West
monsoon and North-East monsoon respectively.
The same concept can be continued by combining the flows in different months, in which case the way to combine
monthly FDCs into an annual FDC is given by:

$$\text{D(q)} = \frac{1}{12}\sum_{m=1}^{12} \text{D}_\text{m}(\text{q}) \tag{10}$$

where $m = 1, \dots, 12$.
If $Q_m$ represents the random variable daily streamflow over $\text{m}^\text{th}$ month, then the expectation $E(Q)$ and variance
$V(Q)$ of annual flow in terms of monthly flows can be expressed as





$$E(Q) = \frac{1}{12} \sum_{m=1}^{12} E(Q_m) \tag{11}$$

$$V(Q) = \frac{1}{12} \sum_{m=1}^{12} E(Q_m^2) - \left(E(Q_A)\right)^2 \tag{12}$$

The relative contributions of Non-monsoon ($C_{NM \to AN}$), South-West monsoon ($C_{SW \to AN}$) and North-East monsoon
($C_{NE \to AN}$) flows to annual flow can be approximated through following expressions:

$$C_{NM \to AN} = \frac{\tau_1 E(Q_{NM})}{\tau_1 E(Q_{NM}) + \tau_2 E(Q_{SW}) + \tau_3 E(Q_{NE})} \tag{13}$$

$$C_{SW \to AN} = \frac{\tau_2 E(Q_{SW})}{\tau_1 E(Q_{NM}) + \tau_2 E(Q_{SW}) + \tau_3 E(Q_{NE})} \tag{14}$$

$$C_{NE \to AN} = \frac{\tau_3 E(Q_{NE})}{\tau_1 E(Q_{NM}) + \tau_2 E(Q_{SW}) + \tau_3 E(Q_{NE})} \tag{15}$$

Similarly, the relative contributions of monthly flows to annual flow can be expressed as:

$$C_{m \to AN} = \frac{\frac{1}{12} E(Q_m)}{\frac{1}{12} \sum_{m=1}^{12} E(Q_m)} \tag{16}$$

where, $m = 1, 2, \ldots, 12$, represents the index for months.
Note, as before, these relative contributions to total flow effectively also measure the relative contributions of the
seasonal/monthly flows to the mean of the annual flow duration curve.
The methodology for constructing annual FDC using seasonal FDC is as follows:
1. The empirical PDFs – $f_{NM}(q), f_{SW}(q)$ and $f_{NE}(q)$ are derived for daily streamflow time series for Non-
monsoon, South-West monsoon and North-East monsoon seasons respectively.
2. These PDFs are then multiplied by scaling factors, $\tau_1, \tau_2$ and $\tau_3$ in equation 9. The scaling factors represent
relative durations of the three seasons considered. For example, $\tau_1 = 5/12$, as the duration of duration of non-
monsoon season is 5 months.
3. The PDF of annual flow is estimated as the weighted sum of three scaled density functions corresponding to
three seasons (see Eq. 7). The annual flow consists of the daily streamflow for Non-monsoon, South-West
monsoon and North-East monsoon seasons.
The performance of the time scale partitioning framework is assessed using the metric, root mean square error
(RMSE). The method of estimation of $q_{\text{sim}}$ is shown in Fig. S3.


$RMSE = \sqrt{\frac{1}{n}\sum_{i=1}^{n}(q_{actual} - q_{sim})^2}$                                                     (17)
**4 Results**
**4.1 Time scale partitioning**
We initially investigated the spatial variations in seasonal and annual flow duration curves across Peninsular India
employing the partitioning framework. The annual flow duration curve and seasonal flow duration curves for
Non-monsoon, South-West monsoon, and North-East monsoon are shown in Fig. 4 for eight representative
gauges, one at the upstream and one at the downstream of each of the four river basins. The estimated annual flow
duration curve (red curve) using the equation 7 is also shown in Fig. 4. Daily streamflow time series is normalized
by catchment area before plotting (on a semi-log paper) the flow duration curve for comparison across the gauging
stations. In particular, the annual flow duration curve (black scatter) is reproduced well by the partitioning of both
seasonal (red curve in Fig. 4) and monthly flows (red curve in Fig. S4). The mean and variance of annual flows
are also reproduced well by the time scale partitioning framework (Fig. S5). This confirms the efficacy of the time
scale partitioning approach of seasonal/monthly flows in approximating the annual flow duration curve (see also
Fig. S4, Fig. S5.a and Fig. S5.d in Supplementary Material).
Another feature that can be observed in Fig. 4 is that in gauging stations located in the northern part of the
peninsular region, flow duration curves (FDCs) of South-West monsoon flows (orange curve) are relatively higher
than other seasonal FDCs. Given the logarithmic scale used to plot of the flows, this dominance is significant. In
sites located in the southern part of the region, the dominance of South-West monsoon is not as strong and North-
East monsoon flows (blue curve) are also significant.
Motivated by these observations, we extracted seasonal and monthly streamflow time series from the entire dataset
across all gauging stations to compute the relative contributions of seasonal and monthly flows to the annual flow
duration curve. The results are presented in Fig. 5. At the monthly scale (top panel, Fig. 5), the contributions of
flows during the months of June to September are much higher than in other months in northern Peninsular basins
(Mahanadi and Godavari, Krishna to a less extent). This can be explained by the contribution of monthly rainfall
to annual rainfall, which is higher during these months as shown in Fig. 6. On the other hand, in the southernmost
Cauvery basin, the dominance of June to September months is relatively not as strong, and there is also a
significant contribution during the months of October to December, higher than in northern basins (Fig. 5.d). This
can be attributed to the slightly more equal dominance of both South-West (June - September) and North-East
(October – December) monsoons over the Cauvery basin (Fig. 6.d) than in the northern basins. This pattern is also
reflected at the seasonal scale (bottom panel, Fig. 5), with the contribution of South-West monsoon flow to annual
flow being slightly higher than that during the other seasons, and much higher in northern basins. However, the
contribution of South-West monsoon to annual flow decreases in southern basins, while the contribution of North-
East monsoon increases, as can be seen clearly in Fig. 5.h for the Cauvery basin. The contribution of Non-monsoon
to annual flow is also higher in southern basins relative to northern basins. This can be attributed to carry over
flows from winter rains during the North-East monsoon, which is more pronounced in the southern part of the
region.

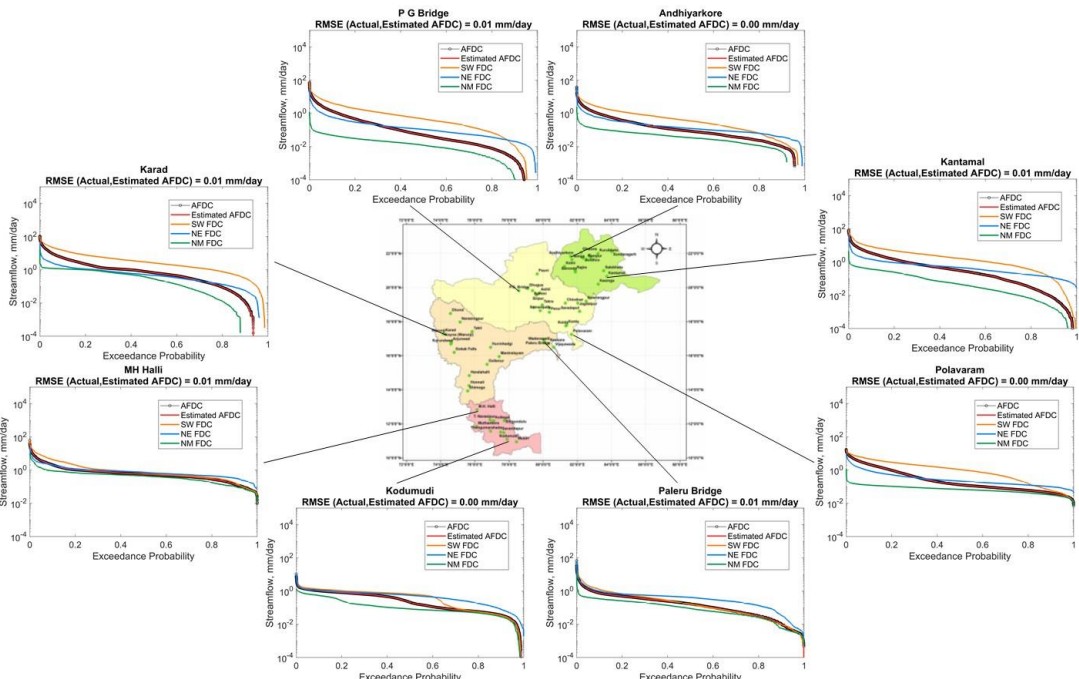


**Figure 4.** Spatial variations in seasonal and annual flow duration curves across Peninsular India. The time scale partitioning framework of seasonal flows in approximating annual flow duration curves works reasonably well.


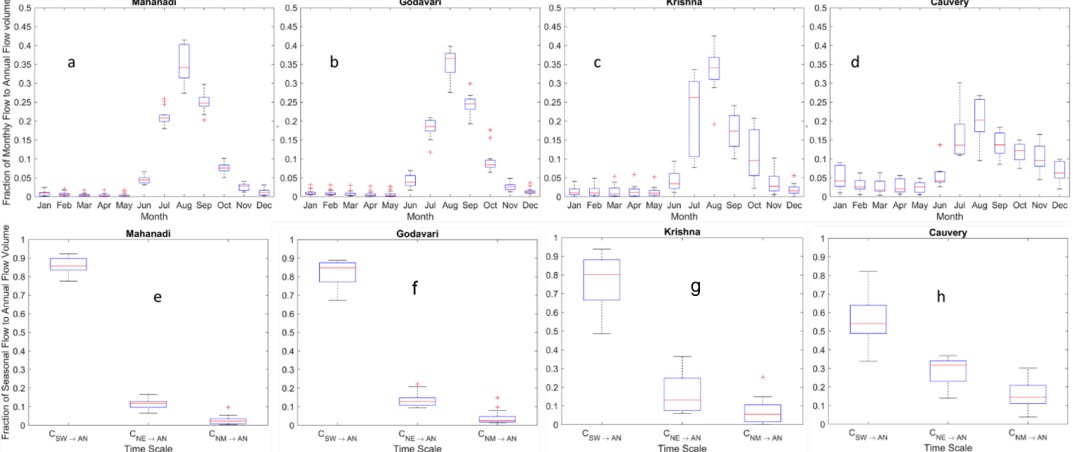


**Figure 5.** The relative contributions of monthly and seasonal flows to annual flow at basin scale. The contributions of South-West monsoon flow to annual flow increases in northern basins whereas it decreases in southern basins. However, the contributions of North-East monsoon flow to annual flow increases towards southern basins.

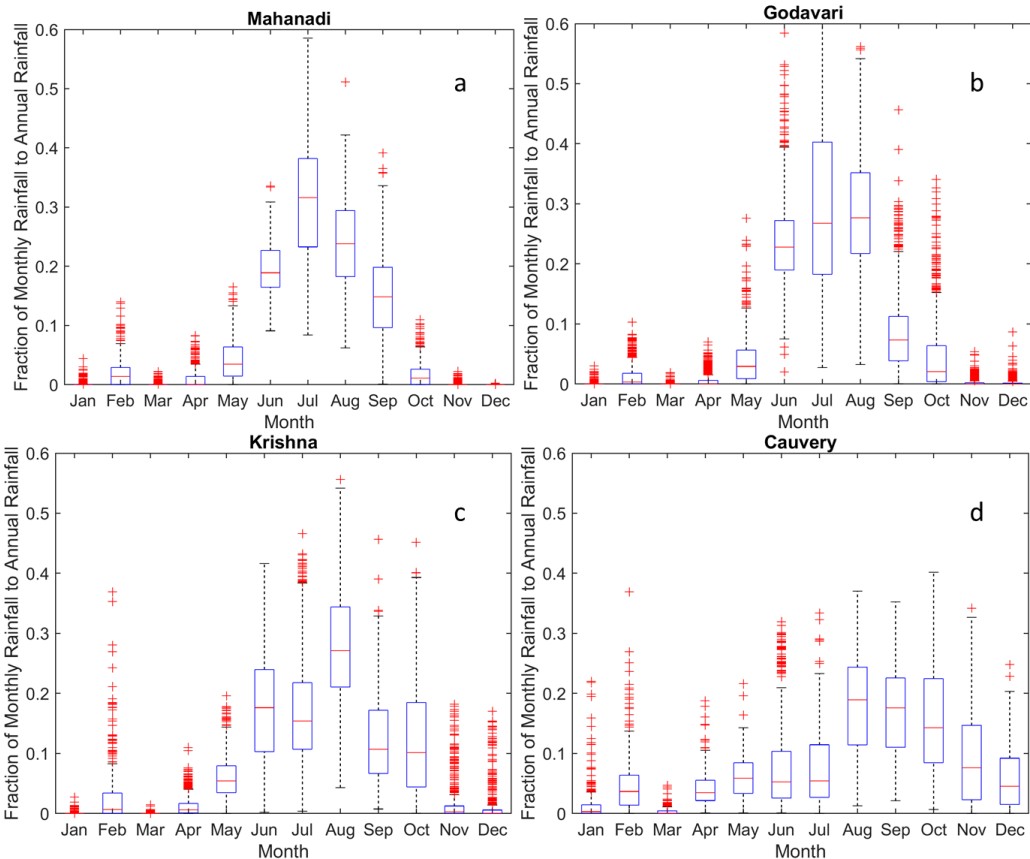


**Figure 6.** Long-term (1951-2010) fractional contribution of monthly rainfall to annual rainfall across Peninsular
basins.





We next carried out regional scale analysis by considering streamflow time series of all the gauging stations across
all four river basins. Similar to basin scale analysis presented before, the relative contributions of seasonal and
monthly flows to annual flow are now estimated at the regional scale (Fig. 7). The spatial patterns of South-West
and North-East monsoon rainfall across the Peninsular region are plotted for comparison using IMD gridded
rainfall product (Fig. 7.b and Fig. 7.e).
The contribution of South-West monsoon flows to annual flow increases in the northerly direction (Fig. 7.a). The
mountainous region of the southern Peninsula (western part of Krishna basin and north-western part of Cauvery
basin) receives high rainfall during the South-West monsoon season (Fig. 7.b – extended till 17° N latitude). The
streamflow produced in the headwater regions of southern basins in response to high rainfall, contributes at least
70% of the annual flow (Fig. 7.a). Yet, the areal fraction of these high rainfall, headwater regions within the four
river basins is quite small and their contributions to the average precipitation or flow at the basin scale is much
smaller. There is also considerable variability in the contributions of South-West monsoon flows to annual flow
in the sub-basins located at the eastern and south-eastern parts of Krishna and Cauvery basins (represented by the
scatter below the regression line till 17° N latitude in Fig. 7.a) due to declining rainfall (Fig. 7c). This considerable
variability, on average, reduces the overall contributions of South-West monsoon to annual flow in southern
Peninsula with respect to the basins in the northern part.
The northern part of the Peninsular region receives comparatively higher rainfall than the southern part without
considering the Western Ghats. This increased rainfall is attributed to the movement of low-pressure systems that
develop over the Bay of Bengal towards central India (Krishnamurthy & Ajayamohan, 2010; Prakash et al., 2015).
The low-pressure systems are a regular feature of South-West monsoon, which brings significant amount of
rainfall in the northern part of the Peninsular region (Krishnamurthy & Ajayamohan, 2010). The increased rainfall
(Fig. 7.b – after 16° N latitude) is responsible for higher contribution of South-West monsoon flows to annual
flow in the northern basins. As the spatial variability of this rainfall is comparatively less than in the southern
Peninsular region, there is less variability in the contribution of South-West monsoon flows to annual flow. The
spatial variability in South-West monsoon along the south-north direction across Peninsular region can explain
the gradient in the contribution of South-West monsoon flows to annual flow in the same direction.
On the other hand, the contribution of North-East monsoon flows to annual flow increases in the southerly
direction (Fig. 7.d and Fig. 7.e). This can be explained by the fact that the southern part of the Peninsular region
receives higher rainfall during North-East monsoon than the rest of the Peninsular region (Fig. 7.f).



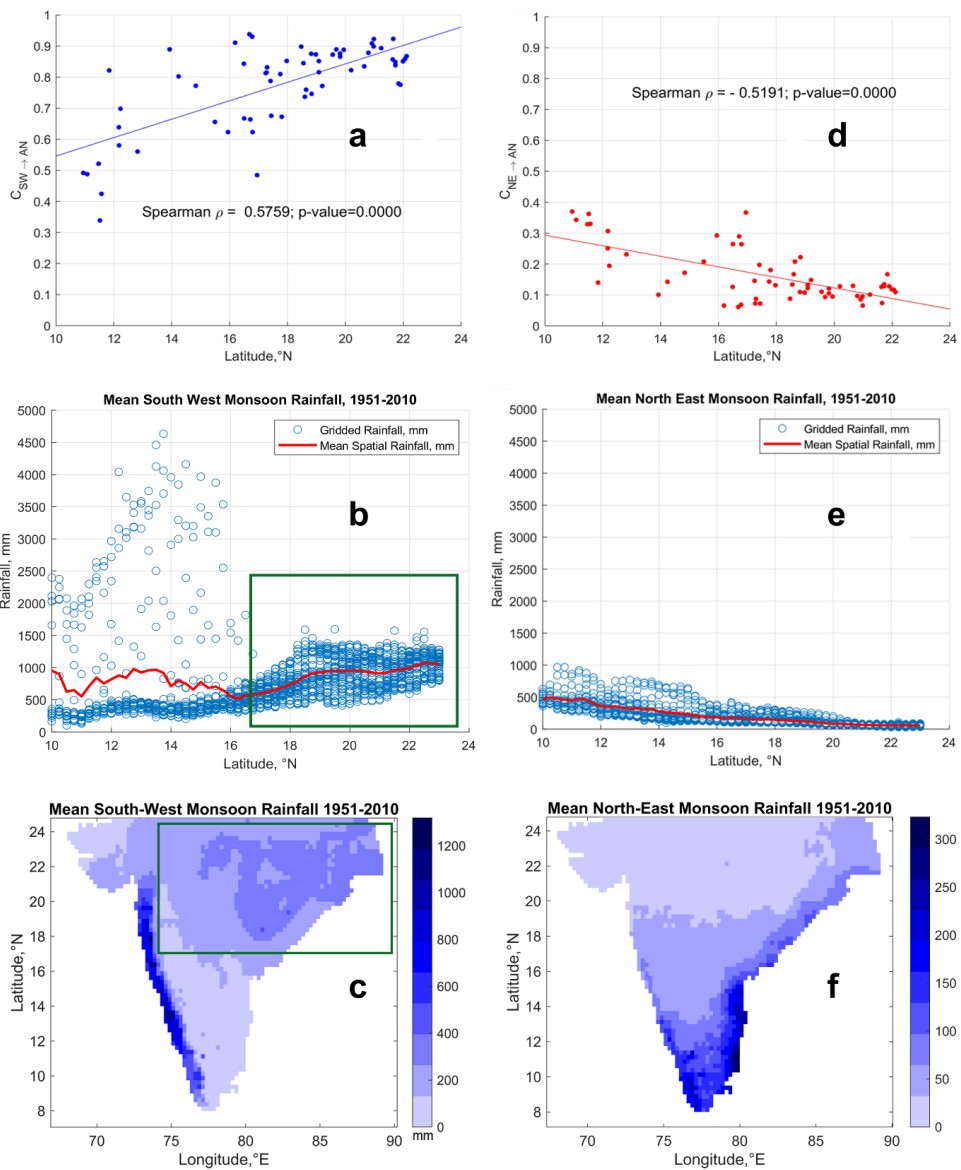

**Figure 7.** Contribution of seasonal flows to annual flow at regional scale. The spatial variability of South-West and North-East monsoons can explain the variation in contributions of seasonal flows to annual flow across south-north gradient. The green box in (b) indicates the northern part of peninsular region which receives higher rainfall than the southern part. The green box in (c) indicates the spatial extent of the rainfall grids which was considered in figure (b). The red line in figure (b) indicates the mean rainfall – obtained by averaging the rainfall values at a specific latitude (˚N).





The application of the analysis framework used here is based on the critical assumption of independence of flows between different seasons (months), which needs to be critically evaluated. Moisture carry-over across seasons is a confounding issue in the case of strongly seasonal catchments (i.e., exhibiting sharp transition from wet season to dry season in terms of rainfall climatology), specifically when the initial wetness condition at the onset of the dry season depends on the final wetness at the end of wet season and vice-versa. Although most of the rainfall (58-90%) is concentrated during South-West monsoon months (i.e., June – September, red bar in Fig. S6) in Peninsular basins, more than 10% of the annual rainfall is received during North-East monsoon months (i.e., October – December, yellow bar for Cauvery and Krishna in Fig. S6). In addition, more than 8% of annual rainfall occurs in non-monsoon season (i.e., January – May, blue bar in Fig. S6. This highlights that rainfall received during non-monsoon and North-East monsoon seasons are comparable, and thus it is difficult to distinguish the rainfall climatology across these seasons. Therefore, it is challenging to declare these are catchments with seasonally dry climates. In order to justify our assumption in the reconstruction of annual FDC from seasonal flows, we have now conducted a multivariate Hoeffding test (Gaißer et al., 2010) to check the independence between three random variables representing Non-monsoon, South-West Monsoon and North-East Monsoon flows respectively. A value of test statistic – $\varphi^2$ – close to zero indicates independence between three random variables. It is observed that except for two stations in Krishna basin, 60 out of 62 stations show independence between flows across the seasons (Fig. S7). This supports appropriateness of the assumption of no carry-over that had been used in this study to construct annual FDC based on seasonal FDCs.

### 4.2 Combined influence of time scale and process scale partitioning

In order to further explore the climatic and landscape controls of streamflow variability regionally, we next partition streamflow into fast and slow flow components, notionally representing surface runoff, and a combination of subsurface and groundwater flow respectively (Ghotbi et al., 2020a, b) (see details in Text S2 and Fig. S8 in Supplementary Material). Fast flow is controlled by event scale runoff generation processes and its variability is characterized by topography, land use, soil and rainfall characteristics. On the other hand, climate seasonality and geologic formations of the subsurface are primary controllers of slow flow variability (Ghotbi et al., 2020a, b). The slow flow component is extracted from observed streamflow by using a recursive digital filter (see details in Appendix A.1). The fast flow component is obtained by then subtracting the slow flow from observed streamflow. The relative contributions of fast flow and slow flow to total flow (and hence also mean annual flow) are estimated using equations S2 and S3 respectively, for all the gauging stations across all four basins. The relative contributions of fast and slow flows to total flow at the basin and regional scales (combining all the gauging stations) are shown in Fig. 8. In addition, the long-term mean annual rainfall across the Peninsular region is also presented for comparison and to possibly explain the contributions of fast flow (Fig. 8.h).

The contributions of fast and slow flows to total flow in each of the four river basins is presented in Fig. 8.a to Fig. 8.d, indicating a strong dominance of fast flow in the northern basins (close to 80% in Mahanadi, Godavari and Krishna), and relatively less dominance (around 60%) in the southern Cauvery basin. This dominance of fast flow also shows up at the regional scale (Fig. 8.e). The regional variations of the relative contributions of slow and fast flows to total flow can also be seen in the results for individual gauges presented in Fig. 8.f and Fig. 8.g, respectively. On average, the contribution of slow flow decreases in the northerly direction, while the contribution of fast flow increases in a corresponding way.



The contribution of fast flow to total flow increases in the northern direction of the Peninsular region (Fig. 8.g).
The fast flow component of streamflow is generally more responsive to the characteristics of rainfall intensity.
The southern part of the region receives high rainfall over Western Ghats along the western edge of Krishna basin
and Cauvery basin (Fig. 8.h). In Cauvery basin, the headwater catchments (namely, MH Halli, Muthankera and
Thengumarahada in Fig. 6) contribute 57 – 65 % of fast flow to total flow locally. The subbasins located at the
western edges of Krishna basin contribute 80% of the fast flow to total flow (between 13° N and 18°N latitudes
in Fig. 8.g) locally. However, there is a wide range of variability in the contributions of fast flow to total flow for
subbasins located in the eastern part of Krishna basin. The spatial mean rainfall increases and variability decreases
after 16° N latitude (Fig. 8.h), which dictate the increased contribution fast flow to total flow. Therefore, the
spatial characteristics (mean and variability) of annual rainfall control the south-north gradient in fast flow
contributions to total flow. In order to explain the variability in slow flow fraction of total flow, a multivariate
regression analysis is performed (details are provided in Appendix, A.5). It is observed that the location of the
gauges is an important predictor of the slow flow fraction of total flow in Peninsular region, revealing the existence
of regional groundwater gradient in the region (Table A.1). In addition to the location of the gauges, the recession
parameter, $\beta$ – that controls the aquifer geometry and water level elevation profile during early and late stages of
recession – is found to be significant in explaining the slow flow fraction of total flow (Table A.1).
The contributions of slow flow to total flow increases in the southerly direction over the Peninsular region (Fig.
8.f). This can be explained by two major factors. Firstly, the Peninsular region is mostly dominated by hard rock
geologic formations, where the subsurface flows are controlled by secondary porosities due to weathering and
fracturing (Chandra, 2018; Das, 2019). The distribution of these formations is highly heterogenous (Fig. 1.c) and
is responsible for baseflow (slow flow) contribution to total flow (Collins et al., 2020; Narasimhan, 2006). For
example, 84% of the total area of Cauvery basin is classified as moderate and good groundwater potential zone
(Arulbalaji et al., 2019). The influence of such potential regions of Cauvery basin is reflected on the presence of
significant amount of slow flow even in the Non-monsoon season (Fig. 9.g and Fig. 9.h). Likewise, 63% of the
total area of Krishna basin is classified under same category (Harini et al., 2018). However, the slow flow regime
becomes much more seasonal (Fig. 9) in the northern part of the Peninsular region due to limited capability of
geologic formations in transmitting slow flow (Patil et al., 2017) as well as strong seasonality in rainfall patterns
(Fig. 9). Secondly, the southern part of the Peninsula receives rainfall almost equally during both South-West and
North-East monsoons, which is reflected in the bimodal pattern of rainfall seasonality (Fig. 9.g and Fig. 9.h). The
compounding effect of bimodal rainfall seasonality and higher fraction of moderate to good groundwater potential
zones explains the higher contribution of slow flow to total flow in southern part of the Peninsular region.



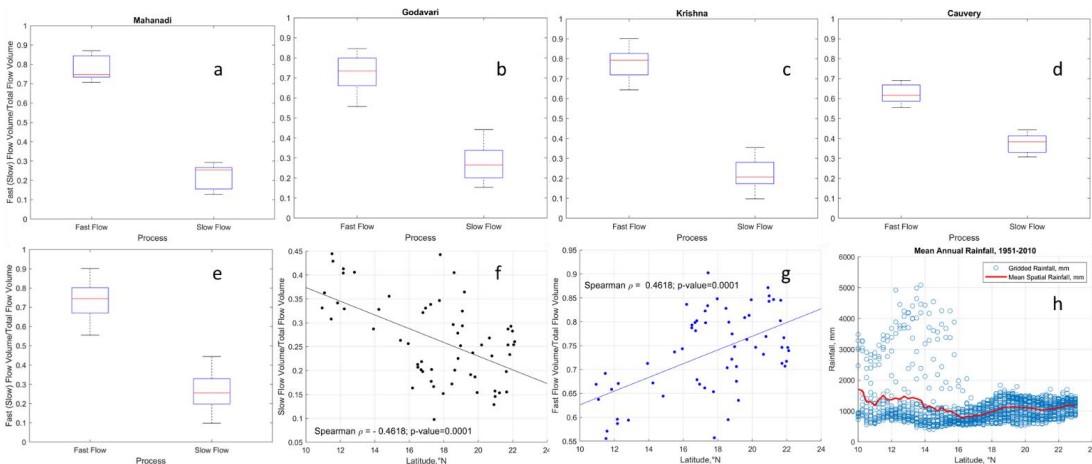


**Figure 8.** Relative contributions of fast and slow flow to total flow. Consistent higher contribution of fast flow
and lower contribution of slow flow to total flow are observed in Peninsular India (a – d) at basin scale. At regional
scale, a systematic gradient in fast and slow flow contributions is observed (f and g). The spatial patterns of rainfall
(h) can explain the gradient in fast flow contributions. The high scatter of rainfall in the low latitudes represents
the heavy rainfall with high variability occurring in the Western Ghats.

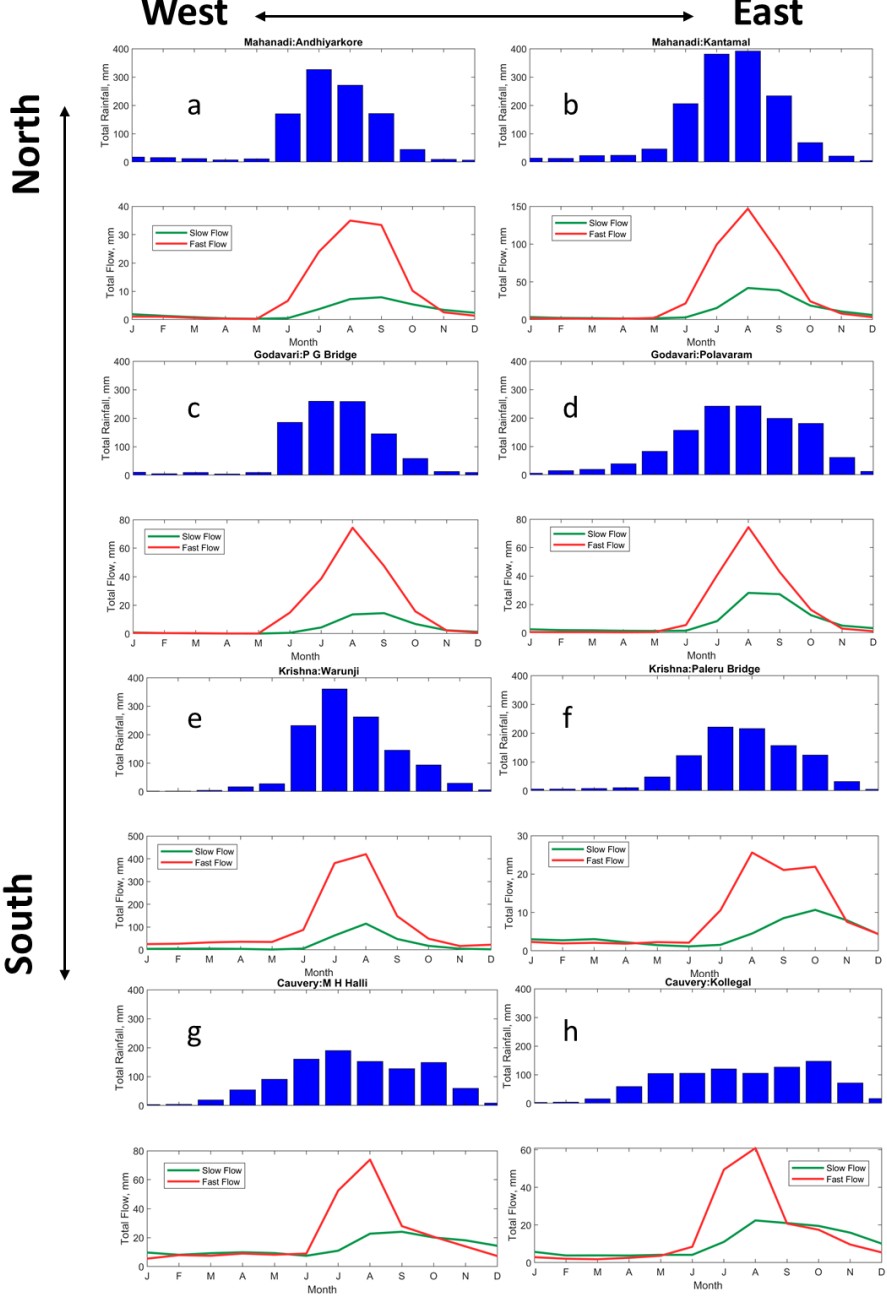

**Figure 9**. Spatial variation of long-term monthly fast and slow flow components of streamflow at selected gauges in Peninsular region. The blue bar plots represent the long-term monthly rainfall averaged over the sub-basins corresponding to the gauging stations. The seasonality in rainfall patterns changes (unimodal to bimodal) across north-south direction of the Peninsular region.



Further, an investigation of the combined influence of climatic time scales and process time scales is therefore pertinent to fully understand the controls of streamflow variability in this region. To address this question, we extracted the fast and slow flow components for each of the Non-monsoon, South-West monsoon and North-East monsoon seasons. These components are then used to estimate their relative contributions to total flow for the three seasons across all the gauging stations.

The relative contributions of fast and slow flow to total flow at basin scale are shown in Fig. 10. It is observed that during the Non-monsoon period, the median contributions of fast and slow flow for Mahanadi, Krishna and Cauvery basins are similar, although there exists considerable variability in their distribution. With the onset of the South-West monsoon, the contribution of fast flow to total flow increases markedly for all the basins, although relatively much less in the Cauvery basin. During the subsequent North-East monsoon season, the contribution of fast flow decreases whereas slow flow contribution increases. The fluctuations in the fast flow contributions can be explained by the onset and withdrawal of the monsoon seasons, which are major contributors to fast flow generation. The fluctuations in the fast flow contributions across seasons can be explained by the differences in the rainfall amount during South-West and North-East monsoons (Fig. 7.c and Fig. 7.f). Among all four basins, the difference in median contributions of fast and slow flow is minimum. These can be attributed to the presence of higher fraction of moderate and good groundwater potential zones (Arulbalaji et al., 2019) which promotes baseflow even in dry periods (Fig. 9.g and Fig. 9.h). The presence of bimodal pattern in rainfall seasonality due to both South-West and North-East monsoons minimizes the difference between the relative contributions of fast and slow flow to total flow.

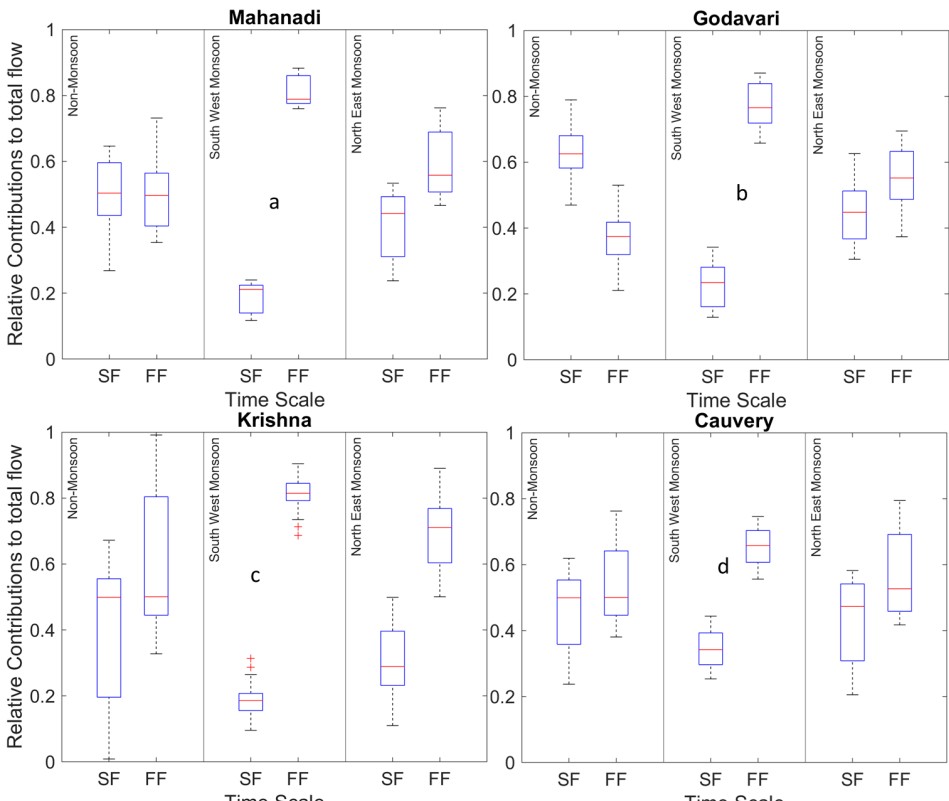

**Figure 10.** Seasonal contributions of fast (FF) and slow flow (SF) to total flow at basin scale.

**5. Validation of stratification of streamflow variability**

**5.1 Understanding physical controls and spatial variation of flow duration curve by fitting statistical distributions**

So far in this paper, in order to understand the physical controls on regional streamflow variability across Peninsular India we have partitioned observed streamflows in two ways: (i) seasonal/monthly flows, and (ii) slow and fast flows. We looked at the relative contributions of these components to mean annual streamflow, looked at how the relative contributions varied regionally, and attributed these to the relative strengths of the monsoons and spatial variations of geological formations. We now return to the FDCs of the flow components, especially the shapes of the FDCs (as reflected in the parameters of the fitted distribution) and look at how they themselves vary regionally.

In our study the fast and slow flow time series are scaled by their respective long-term mean values to remove the influence of mean climate and geology, thus providing an opportunity to identify the secondary controls on the variation of shapes of FDCs. The scaled fast and slow flow time series are now used to fit the mixed gamma distribution (MGD, (see details in Appendix A4). The parameters of mixed gamma distribution control the shape





and orientation of the FDC. For example, the shape parameter $k$ controls the slope of the FDC whereas $\alpha$ controls
the zero-flow part of the FDC. However, the parameter $\theta$ affects the vertical shift of the FDC. In addition, these
parameters are also linked with the mean and variance of the streamflow time series. For example, the scale
parameter $\theta$ is directly proportional to the mean of the time series whereas, the shape parameter $k$ is inversely
proportional to the variance of the time series.
As the fast and slow flow time series are scaled with their respective long-term means, the scale parameter $(\theta)$ is
approximately found to be inversely proportional to shape parameter $(k)$ through the relationship, $k\theta = \frac{1}{1-\alpha}$
(Cheng et al., 2012). Therefore, the variations of only $k$ and $\alpha$ – zero-flow probability, are presented in this section.
The variation of $k$ can be related to the steepness of the FDC, i.e., smaller values of $k$ will have steeper slopes.
The Nash-Sutcliffe efficiency (NS) and coefficient of determination ($R^2$) goodness of fit of fast/slow flows to
MGD is shown in Fig. S10 (in Supplementary Information). In addition, the observed and simulated fast and slow
flow FDCs are compared in Fig. S8 (in Supplementary Information). It is observed that the slow flow component
fits well to mixed gamma distribution than fast flow component, as slow flow is most stable component and MGD
satisfactorily captured the shape of slow flow FDC. However, MGD adequately captures the shape of fast flow
FDCs at upper tail (high flow segment), except for the lower tail (low flow segment). The fast flow processes are
governed by more complex processes (for example, infiltration and saturation excess runoff generation, runoff
routing, stochastic nature of storm events, properties of soil and topography etc.) than slow flow (for example,
climate seasonality and underlying geology of aquifer system).
The seasonal variation of parameters of the mixed gamma distribution at regional scale (comprising of all the
gauging stations) is presented in Fig. 11. The mixed gamma distribution performed well in fitting the flow duration
curves of two flow components across different seasons (Fig. S10). In Fig. 11.a, it is observed that the shape
parameter of slow flow is consistently higher than that of fast flow. The shape parameter is inversely proportional
to the variance of streamflow. The slow flow exhibits lower variance due to its longer time of residence in the
subsurface formations. Moreover, the subsurface formations in Cauvery River basin are more favourable to slow
flow in comparison to the other three basins (Fig. 9.g and Fig. 9.h). In addition, the bimodal seasonal pattern of
rainfall is also responsible for occurrence of slow flow even in the Non-monsoon period for the southern basins
(Fig. 9).

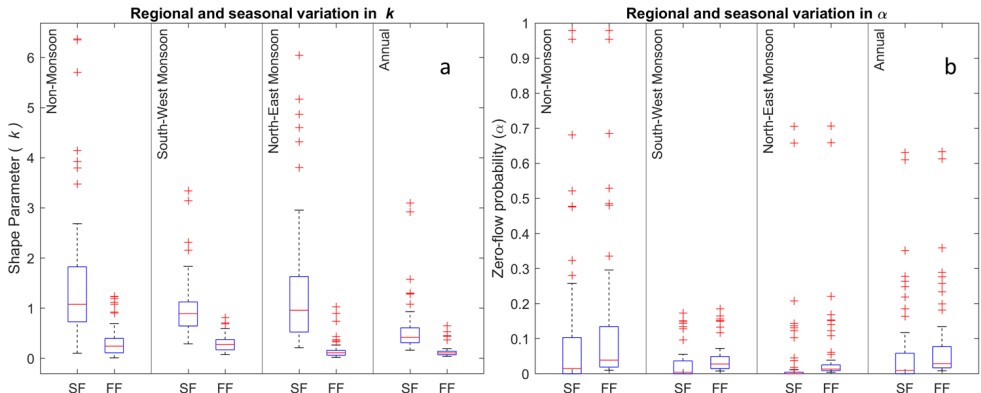

**Figure 11.** Regional and seasonal variation of $k$ and $\alpha$ parameter of mixed gamma distribution.

The fast flow component exhibits higher variance than the slow flow component. The median shape parameter of fast flow is highest during South-West monsoon season and lowest during North-East monsoon (Fig. 11.a). This can be explained by the lower variance of fast flow during South-West monsoon as the rainfall amount is higher during the season compared to the North-East monsoon (Fig. 7.c and Fig. 7.f). The dominance of both South-West and North-East monsoons in Cauvery basin results in lower variance of fast flow compared to the northern basins. The fast flow duration curves are steeper than the slow flow duration curves for all seasons, as the magnitudes of $k$ for fast flow are smaller than that of slow flow (Fig. 11.a).

The parameter $\alpha$ controls the zero-flow part of the flow duration curve. It is observed that the mean $\alpha$ for slow flow is minimum during South-West monsoon and maximum for Non-monsoon season (Fig. 11.b) on a regional scale. This can be attributed to the combined influence of rainfall during South-West monsoon and the connectivity between underlying geologic formations in the Peninsular region. For the fast flow, the mean $\alpha$ is minimum during the South-West monsoon and maximum during Non-monsoon as the South-West monsoon is the dominating rainfall season in Peninsular India.

The shape parameters ($k$) of MGD for slow and fast flow components are linked with landscape properties through recession analysis, where the parameters $\gamma$ & $\beta$ of power-law relationship are estimated using streamflow data (details in Appendix A.2). It is observed that shape parameter (inversely proportional to variability) of slow flow is positively correlated with $\beta$. The parameter $\beta$ is influenced by aquifer geometry and water table elevation profile defining early and late stages of recession (Tashie et al., 2020a; Tashie et al., 2020b). Higher values of $\beta$ indicate slow late recessions which is characterized by low variability in slow flow (Fig. 12.a).

The shape parameter of fast flow is negatively correlated with the parameter $\gamma$ of the power-law relationship (Fig. 12.b). The parameter $\gamma$ strongly related with the seasonality of catchment wetness and evapotranspiration which are primary governing factors for runoff generation (Dralle et al., 2015; Gnann et al., 2021). In addition, the spatial variation of rainfall also influences the variability of $\gamma$ (Biswal & Kumar, 2014) which reflects the variability of fast flow.





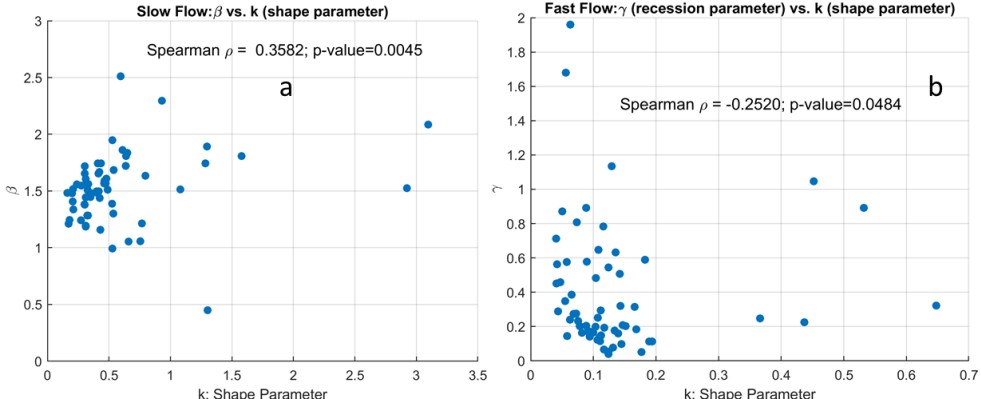

**Figure 12.** Relationship between flow variability (related inversely to shape parameter, k of mixed gamma distribution) and recession parameters.

The variation of the parameters, $k$ and $\alpha$ was also studied using spatial descriptors (latitude and longitude) as explanatory variables to understand the spatial variation of FDCs across south-north, west-east gradients. In addition, the behaviour of these parameters is also assessed using catchment area as another explanatory variable. The regional parameter sets comprising of $k$ and α are next constructed for slow and fast flow processes by including these parameters for all the time series across different gauging stations across the Peninsular region. The Spearman correlation coefficients between these parameters and explanatory variables (i.e., catchment area and spatial descriptors – latitude and longitude) for slow and fast flow processes at seasonal scales are computed. The schematic representation of significant directions (positive/negative correlations) in Spearman coefficient is shown in Fig. 13.


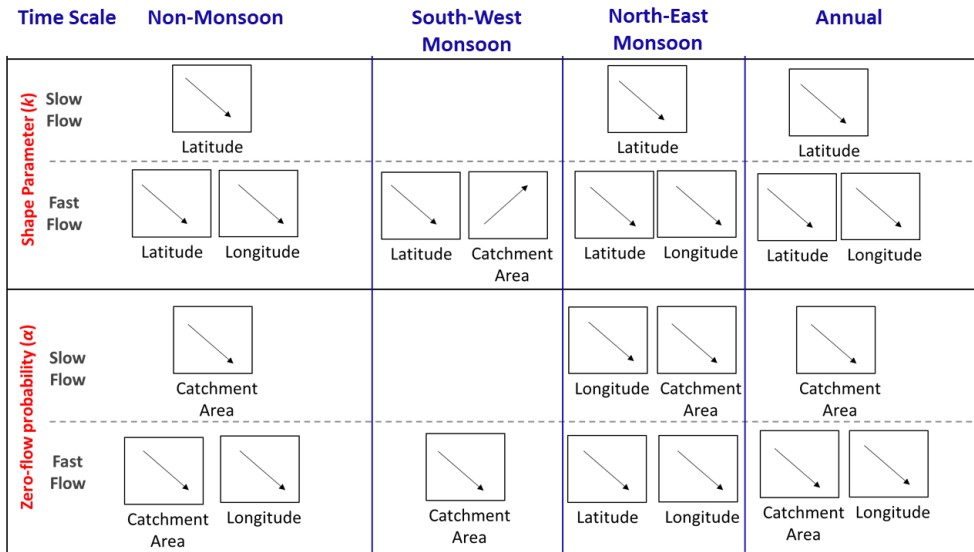

**Figure 13.** Schematic representation of spatial and temporal variation of parameters of mixed gamma distribution across Peninsular India. The direction of significant Spearman correlation coefficient between model parameters and descriptors (catchment area and spatial descriptors – latitude and longitude) for fast and slow flow across multiple time scale is presented.

The shape parameter of fast flow is found to be positively correlated with catchment area (Fig. 13, top panel), implying lower variability of fast flow in large catchments. This can be attributed to increased smoothening effect of incoming rainfall in larger catchments through various storages, thus reducing the variability of fast flow. Moreover, the shape parameters for fast flow are negatively correlated with spatial descriptors, indicating increased variability of fast flow along south-north and west-east gradients. This can be partly explained by the bimodal seasonal pattern of rainfall due to dominance of South-West and North-East monsoons, thus reducing the variability of fast flow in the southern part of the region. The rainfall pattern becomes more seasonal (primarily due to South-West monsoon) in the northern part of region which can contribute to increased variability of fast flow. The presence of numerous water retention structures for supporting irrigation in these regions (54 – 75% of Peninsular basins are crop land) can modify the variability of the flow, although we have not analysed this separately in this study.

The shape parameter of slow flow is found to be negatively correlated with latitude, implying that slow flow becomes highly variable in the northern part of the region. This can be explained by the nature of geologic formations in the Cauvery basin that promotes slow flow even during the Non-monsoon period. However, in the northern part of the region, the slow flow tends to become more seasonal and has very limited flow during non-rainy seasons. In addition to the geology, the bimodal seasonal rainfall patterns due to monsoons can play an important role in the variability of slow flow. Apart from the spatial descriptors, the slow flow variability is inversely proportional to catchment area, implying larger catchments have lower slow flow variability than smaller catchments. This can be explained by the proportional increase in area of contribution to slow flow with increase in catchment size, thus reducing the variability in slow flow for larger catchments.





The parameter $\alpha$ is found to be negatively correlated with catchment area (Fig. 13, bottom panel) for fast and slow
processes, implying zero-flow probabilities are lower for larger catchments. The higher residence time of water
in larger catchment due to various kinds of storages facilitates flow in river even in Non-monsoon season, thus
reducing the zero-flow probabilities. In addition, the parameter $\alpha$ of both slow and fast flow are negatively
correlated with longitude, implying lower zero-flow probabilities along west-east direction. This can be attributed
to natural declining elevation (Fig. S1.b) which promotes both fast and slow flow towards eastern direction.
**5.2 Understanding physical controls and spatial variation of seasonal flow duration curve using mid-section**
**slope**
Apart from mean, variance and no-flow frequency, the midsection slope of the FDC – estimated using
$\frac{\ln(Q_{33p})-\ln(Q_{66p})}{0.66-0.33}$, where $Q_{33p}$ and $Q_{66p}$ represent the streamflow values at 33rd and 66th percentiles respectively –
is connected to the average flow regime of the catchment, which is controlled by both surface and subsurface
processes (Yokoo & Sivapalan, 2011; Chouaib et al., 2018). The association of the slope of FDC with the
parameters pertaining to recession analysis is presented in Fig. 14.

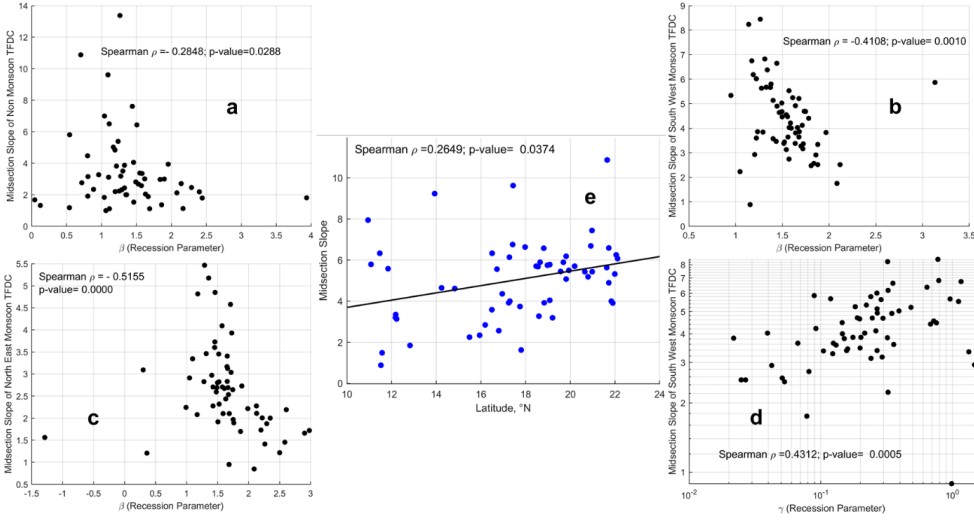


**Figure 14.** Association between streamflow variability and recession parameters.
During Non-monsoon and North East monsoon seasons (Fig. 14a and Fig. 14c) – when rainfall is comparatively
less than South West monsoon – a significant association between flow variability and $\beta$ highlights the importance
of slow flow and recession characteristics controlled by aquifer geometry and water table elevation profile. In
addition to significant association with $\beta$ during South West monsoon (Fig. 14b), the midsection slope of FDC is
positively correlated with $\gamma$ – the parameter which is strongly related with the seasonality of catchment wetness,
evapotranspiration and spatial variation in rainfall – revealing the importance of land surface processes in
variability of streamflow variability.



A coherent pattern in variability of streamflow (via. Midsection slope of FDC) is observed across South – North
gradient of the Peninsular region (Fig. 14e). This systematic pattern in streamflow variability reflects the influence
of combined functioning of subsurface and land surface processes on regional hydrologic signatures of Peninsular
India.
**6. Conclusions**
Being a signature of a catchment's hydrological behavior and a concise graphical summary of streamflow
variability at a specific gauging station, FDC relates the frequency and magnitude of observed streamflows and
helps explain flooding mechanisms and low flow conditions at the referred location. Furthermore, at the catchment
scale, FDCs incorporate the forcing mechanisms of the water cycle and the physical and morphological properties
of the river basin that influence the water partition between surface runoff and baseflow and, thus, control flow
regimes (Costa & Fernandes, 2021). Motivated by this fact, in this study we outlined a framework and its
suitability for understanding process controls of FDCs, which involved separating annual streamflow into seasonal
flow components and constructing annual FDC using seasonal FDCs. The goal of this study was to demonstrate
the efficacy of the framework to explore the process controls on streamflow variability across Peninsular India.
The study followed a data-based approach using streamflow data taken from 62 stream gauges distributed within
four major river basins in Peninsular India. The probability density functions are initially derived for daily
streamflow time series for Non-monsoon, South-West monsoon, and North-East monsoon seasons. These PDFs
are then multiplied by scaling factors that represent relative durations of the seasons considered. The probability
density function of annual flow is then estimated as the weighted sum of three scaled density functions
corresponding to three seasons. The performance of the time scale partitioning framework is then further assessed
using the metric root mean square error.
Analysis and interpretation of the results of the study revealed that the main drivers of regional variability of
streamflow across Peninsular India include (1) major mountain ranges – the Western and Eastern Ghats – which
govern regional atmospheric circulation and precipitation variability; (2) the South-West and North-East
monsoons that occur in different times of the year and come from different directions; and (3) east-west and north-
south gradients of geology. The combined influence of seasonal rainfall patterns, catchment size and the ability
of the subsurface formations to transmit slow flow controls the shape of flow duration curves of the flow
components along south-north and west-east directions in Peninsular region.
To summarize, the major findings of the study are outlined below:
I.    Spatial variations of seasonal and annual flow duration curves across Peninsular India are initially
investigated by approximating the annual flow duration curve via partitioned seasonal and monthly flow
duration curves. FDCs of South-West monsoon flows are relatively dominant to other seasonal FDCs at
stations in the northern portion of the peninsula. From June to September, flow contributions in northern
Peninsular basins are significantly higher than in other months (Mahanadi and Godavari, Krishna to a
lesser extent). However, the contribution from June to September is not as substantial in the southernmost
Cauvery basin; there is also a major contribution from October to December. This is attributable to the
fact that the South-West and North-East monsoons both impact the Cauvery basin. It is further noticed
that the contribution of the North-East monsoon to annual flow is larger in southern basins than in





northern basins. The contribution of the Non-monsoon to annual flow is also stronger in the southern
basin and is attributed to winter rains from the North-East monsoon, which are more evident in the
southern part of the peninsula, creating carryover flows.
II.   The streamflow produced in the headwater regions of southern basins, which extends until 17° N latitude
and contributes at least 70% of the annual flow, is a result of high rainfall during the South-West monsoon
season in the mountainous region of the southern Peninsula (western part of Krishna basin and north-
western part of Cauvery basin). The northern part of the Peninsular region experiences notably higher
rainfall than the southern part, not considering the Western Ghats region. The low-pressure system, which
is a regular feature of the South-West monsoon that brings significant rainfall in the northern part of the
Peninsular region, attributes the increased rainfall (after 16° N latitude) and is responsible for the higher
contribution of South-West monsoon flows to annual flow in the northern basins. The spatial variation in
the contribution of South-West monsoon flows to annual flow in the south-north direction is thus
explained by the spatial variability of the South-West monsoon in the same direction over the Peninsular
region. The contribution of North-East monsoon flows to annual flow, on the other hand, increases in a
southerly direction, which can be explained by the fact that the southern part of the Peninsular region
receives more rainfall during the North-East monsoon than the rest of the Peninsular region.
III.   Spatial variations of fast/slow and total flow duration curves across Peninsular India are then explored
by approximating the total flow duration curve by partitioned flow duration curves. Relative
contributions of fast and slow flows to total flow in each of the four river basins show a significant
dominance of fast flow in the northern basins, close to 80% in Mahanadi, Godavari, and Krishna river
basins.
IV.   The Western Ghats, which run along the western boundary of the Krishna and Cauvery basins, bring a
lot of rain to the southern part of the region. As a result, the western margins of the sub-basins along the
Krishna basin contribute 80 percent of the fast flow to total flow (between 13° N and 18°N latitudes).
However, the south-north gradient in fast flow contributions to total flow is governed by increasing
spatial mean characteristics of annual rainfall after 16° N latitude, which dictates an increased
contribution of fast flow to total flow.
V.   The greater contribution of slow flow to total flow in the southern Peninsular region, particularly Cauvery
and Krishna, is characterized by bimodal rainfall seasonality and the presence of a higher fraction of
moderate to good groundwater potential zones and is responsible for the spatial variation of
increased relative contributions of slow flow to total flow in the southerly direction over the Peninsular
region.
VI.   A coherent pattern in streamflow variability across the South-North gradient of the Peninsular region is
observed via the midsection slope of FDC. These similar spatial variation in streamflow variability
demonstrate the impact of combined subsurface and land surface processes on Peninsular India's regional
hydrologic signatures.
Previous data-based explorations of process controls on the FDC have typically followed a Darwinian (Harman
and Troch, 2014) comparative hydrology approach. They have looked at between-catchment and regional
variations of the FDC (or of parameters of statistical distributions fitted to empirical FDCs), their attribution to
climatic and landscape properties, and their interpretation in terms of their underlying process controls (fast flow





668 and slow flow etc). In the Darwinian approach, each catchment is deemed a particular but statistically independent

669 realization of the coevolution of climate and landscape properties, with the hydrologic response being both a cause

670 and effect in this coevolution (Wagener et al., 2013). The novelty of the data-based exploration of process controls

671 on the FDC adopted in this study is that here we have followed a Wegenerian (cf. Alfred Wegener, Sivapalan,

672 2018) comparative hydrology approach, in which the focus was on exploration of the controls of common regional

673 landscape features (in space) and seasonal climatic variations (in time) features on regional variations of the FDC.

674 We interpret the imprints of the regional variations streamflow variability of the FDCs outlined as findings across

675 Peninsular India as the consequence of several episodes of tectonic, geological, and volcanic activities in the

676 Indian subcontinent ever since the breakup of Gondwana and its collision with Asia during the Jurassic age,

677 resulting in the uplift of mountain ranges, including the Himalayas, and their role in the establishment of India's

678 monsoon climate.

679 We acknowledge, however, that in recent times streamflow variability in Peninsular India has been significantly

680 impacted by anthropogenic activities, including significant land use and land cover changes, and other human

681 interferences such as the building of dams and the extraction of water from both rivers and from groundwater

682 aquifers for human use. The present study has not explored the effects of human impacts: their impacts on both

683 temporal (inter-decadal) and spatial (regional) variations of the FDCs is left for future work. Further work is also

684 needed to understand in more detail the causes and the relative contributions of regional patterns precipitation and

685 geological formations on streamflow partitioning.

686 On the methodological front, there is opportunity to refine the analysis used here to incorporate the statistical

687 cross-correlation between fast and slow flows in the reconstruction of the FDC for total streamflow, by adopting

688 generalized approaches (e.g., copulas). In the exploration of the relative contributions of the monsoons, there is

689 scope to extend the analysis framework to partition the streamflow variability guided by the actual breakdown

690 into the seasons each year in a more flexible way, as opposed to the static way. This is likely to make the results

691 of the analysis more robust and less uncertain. Finally, in the process domain, the filter-based separation of total

692 streamflow into fast and slow flow can be variably impacted by catchment size, introducing some uncertainty into

693 the partitioning of the FDC of total streamflow into its fast flow and slow flow components. Future work in this

694 area should explore ways to overcome these methodological shortcomings.

695

696 **Appendix**

697 **A.1 Baseflow decomposition (Recursive Digital Filter)**

698 The partitioning of total flow ($Q$) into slow flow ($Q_s$) is performed using recursive digital filter technique as

699 described in Arnold & Allen (1999) and Arnold et al. (1995). Based on the study by Nathan and McMahon (1990),

700 they found that a coefficient range between 0.9 and 0.95 yielded most acceptable baseflow separation. Therefore,

701 we have taken the value 0.95 as a coefficient value for this analysis (more discussion is provided at the end of

702 A.1). This filter is applied to daily streamflow timeseries data for all the gauging stations across the Peninsular

703 region.

704 The equation of the filter is





$$q_t = \varepsilon q_{t-1} + \frac{(1+\varepsilon)}{2}(Q_t - Q_{t-1}) \tag{A.1}$$
where $q_t$ is the filtered surface runoff (quick response) at the $t$ time step, $Q$ is the original streamflow (total flow),
and $\varepsilon$ is the filter parameter (which is assumed to be 0.95). Slow flow, $Q_s$, is calculated with the equation:
$$Q_s = Q - q_t \tag{A.2}$$
After obtaining the slow flow component, the fast flow $(Q_f)$ is obtained by subtracting $Q_s$ from $Q$.
$$Q_f = Q - Q_s \tag{A.3}$$
In order to demystify the role of different values of the filter parameter in the digital recursive filter, the model
was run for three different seasons for all the catchments in Peninsular region. The results are presented in Figure
A1.

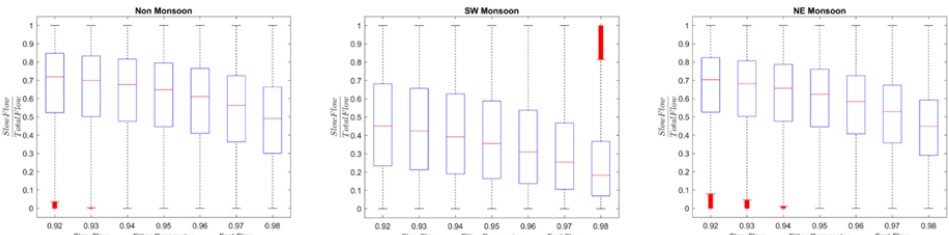


**Figure A1.** Contribution of slow flow to total flow for different seasons. The box plots in each season represent
the partitioning of total flow into slow flow for different filter parameters, viz. [0.92,0.93,0.94,0.95,0.96,0.97
,0.98].

It is observed that the median variations in the slow flow fraction during non-monsoon period (0.5-0.7), south
west monsoon period (0.18-0.45) and north east monsoon period (0.44-0.7) which lies within 30% variation.
However, even with these variations, the overall pattern, i.e., high slow flow contribution during non-monsoon
and north east monsoon seasons and low slow flow contribution during south west monsoon remains intact,
revealing seasonal changes in the dynamics of slow flow contribution to total flow. In this paper, we assumed the
parameter 0.95 reflecting the average variability in slow flow contributions to total flow.
**A.2 Recession Analysis**
In recession analysis, it is often assumed that rate of change of streamflow $\frac{dQ}{dt}$ and streamflow $(Q)$ follows a
power law in the form:
$$-\frac{dQ}{dt} = \gamma Q^\beta \tag{A.4}$$
The parameter $\gamma$ is function of static watershed properties (i.e., hydrological conductivity, drainable porosity,
aquifer depth, aquifer breadth, impermeable layer slope and length of stream) (Tashie et al., 2020a). The parameter
$\beta$ represents the geometry of the contributing aquifer and water table elevation profile that defines the early and
late periods of recession (Tashie et al., 2020b). $\frac{dQ}{dt}$ is estimated using exponential time stepping scheme (Roques
et al., 2017). Strictly decreasing recession segments ($\frac{dQ}{dt} < 0$) with recession segments more than 5 days are
considered for the estimation of the parameters ($\gamma \text{ and } \beta$) (Jachens et al., 2020). A weighted least square
regression is used to fit a line in log-log space to recession segments (Roques et al., 2017). The median of the
parameters is used to describe catchment-average recession behaviour (Gnann et al., 2021).



**A.3 Absolute contributions of fast and slow flow to total flow**

The absolute contributions of fast and slow flow to total flow are determined using the coefficient of determination ($R^2$) of simple linear regression models, that measures the reduction in variability of total flow due to fast and slow flow components. The details are given below:

Model 1: $Q = \varphi_1 \cdot Q_f + \epsilon_1$               (A.5)

Model 2: $Q = \varphi_2 \cdot Q_s + \epsilon_2$               (A.6)

The coefficient of determination measures the effect of slow(fast) flow in reducing the variation in total flow based on Model1(Model2). Higher the value of this coefficient, higher the contribution of slow/fast flow in reducing the variation in total flow.

The coefficient of determinations for two models can be estimated as:

$$R^2_{(1)} = \frac{SSR^{(1)}}{SSTO}$$               (A.7)

$$R^2_{(2)} = \frac{SSR^{(2)}}{SSTO}$$               (A.8)

where, $SSR^{(1)}$ and $SSR^{(2)}$ represent the regression sum of squares for Model 1 and Model 2 respectively, and SSTO represents the total sum of squared deviations from mean, i.e., $SSTO = \sum(Q_i - \bar{Q})^2$. The sum of squares due to the models are expressed as:

$SSR^{(1)} = \sum(\widehat{Q_{(1)}} - \bar{Q})^2$ and $SSR^{(2)} = \sum(\widehat{Q_{(2)}} - \bar{Q})^2$ where $\widehat{Q_{(1)}}$ $and$ $\widehat{Q_{(2)}}$ are the fitted values of total flow using Model 1 and Model 2 respectively.

The values of coefficient of determination ($R^2$) for three seasons are shown in Fig. A2.

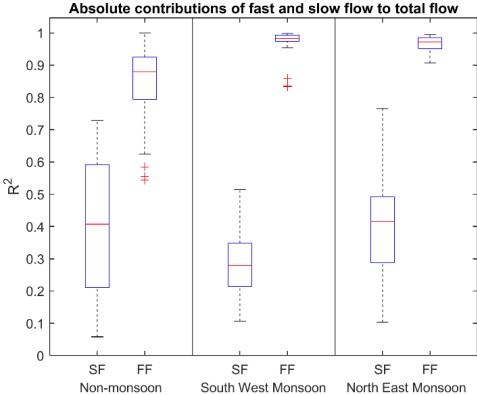

**Figure A2.** Coefficient of determination representing the absolute contribution of fast/slow flow in reducing the variation in total flow across seasons.

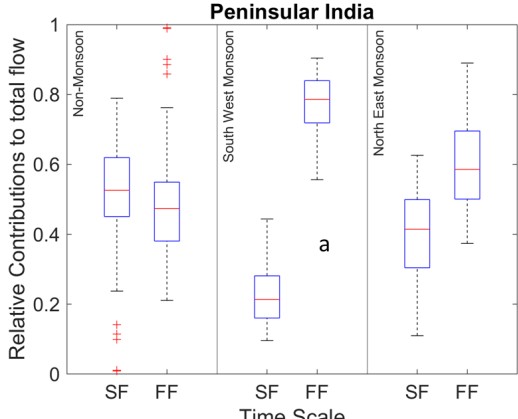

758

**Figure A3.** Relative contributions of fast (FF) and slow flow (SF) to total flow at regional and seasonal scales
(NM – Non-monsoon, SW – South-West monsoon and NE – North-East monsoon).

It can be shown that the pattern of absolute contribution remains similar (in terms of phase relationship between
slow and fast flow contributions to total flow) with that of relative contribution as reported in Fig A3. However,
there are differences in the magnitudes of the absolute contributions and relative contributions of the flow
components to total flow. The major difference between relative and absolute contribution analyses is that the
contribution of the fast flow is significantly higher than the slow flow for non-monsoon season, which can be
attributed to rainfall during the non-monsoon period (Fig 6).

**A.4 Fitting statistical distributions**

A simple statistical distribution, the mixed gamma distribution, is employed here to characterize the FDC in
Peninsular River system. The choice of the mixed gamma distribution is made to take care of the flow regimes of
the selected basins (i.e., to accommodate the presence of zero flow values) (Cheng et al., 2012). The classic gamma
distribution is a two-parameter, continuous distribution with a shape parameter, $k$, and a scale parameter, $\theta$. In
addition, the probability of zero flows, $\alpha$, is defined as the ratio of the number of zero flow days to the total number
of days within the data record. The mixed gamma distribution (Cheng et al., 2012) employed to model FDC is as
follows:

$$f\,(q,k,\theta,\alpha) = \begin{cases} \alpha, & q = 0 \\ (1-\alpha).\,g(q,k,\theta), & q > 0 \end{cases} \qquad (A.9)$$

where $g(q,k,\theta)$ is the probability density function of the gamma distribution. The probability density function of
the gamma distribution is assumed to take the form of (Cheng et al., 2012):

$$g\,(q,k,\theta) = \frac{1}{|\theta|\,\Gamma\,(k)} \left(\frac{q}{\theta}\right)^{k-1} \exp\left(-\frac{q}{\theta}\right) \qquad (A.10)$$





where $k$ and $\theta$ are the shape and scale parameters, respectively. The parameters $k$ and $\theta$ can be estimated by the
method of moments. The mean, μ, and variance, ν, of the gamma distribution are evaluated from the $q > 0$ time
series. The parameters are related to μ and $\nu$ as follows:

$$\mu = k \cdot \theta \tag{A.11}$$

$$\nu = k \cdot \theta^2 \tag{A.12}$$

The following formulation is used to obtain the flow given a probability of exceedance, $p$ (Cheng et al., 2012):

$$q\,(p, k, \theta, \alpha) = \begin{cases} G^{-1}\left(1 - \dfrac{p}{1-\alpha}, k, \theta\right), & 0 \le p \le 1 - \alpha \\ 0 & , \quad 1 - \alpha < p \le 1 \end{cases} \tag{A.13}$$

where $G^{-1}$ is the inverse of the CDF of the mixed gamma distribution.
In this case, given that we have already looked at the climatic and landscape controls on the mean annual flows,
we instead work with the normalized daily streamflow time series (i.e., daily streamflow divided by long-term
mean daily streamflow), which is then used to estimate the parameters of the mixed gamma distribution. The
parameters estimated from the normalized streamflow series can thus be used to infer secondary controls on the
shape of flow duration curves.

**A.5 Investigating the slow flow fraction of total flow in Peninsular India**
The variability in slow flow fraction (SFF) is investigated using multiple linear regression by considering the
recession parameters, $\beta$ and $\gamma$ in the equation $-\frac{dQ}{dt} = \gamma Q^\beta$ and the location of the gauge ($\delta$, latitude). The results
are provided below:
*Regression Model:*

$$SFF = \alpha_0 + \alpha_1 \gamma + \alpha_2 \beta + \alpha_3 \delta \tag{A.14}$$


**Table A.1** – Statistical Assessment of regression coefficients

| Coefficients | Estimate | SE | tStat | pValue |
|---|---|---|---|---|
| $\alpha_0$, (Intercept) | 0.35361 | 0.055275 | 6.3973 | 2.99E-08 |
| $\alpha_1$ | -0.024117 | 0.021119 | -1.142 | 0.25816 |
| $\alpha_2$ | 0.12791 | 0.025704 | 4.9764 | 6.12E-06 |
| $\alpha_3$ | -0.015556 | 0.0023978 | -6.4875 | 2.12E-08 |


The above regression model was able to explain to about 52% of the variability in slow flow fraction of total flow
(p-value = $1.98 \times 10^{-9}$), and in general, the model is found to be useful to explain SFF in terms of recession
parameter and latitude. A fraction of the unexplainable part in SFF can be attributed to the heterogeneity in
subsurface geologic formations and dam induced variations in the catchment storages. However, at a regional
scale, the south-north gradient (represented by the parameter $\delta$) can explain the variability in slow flow fraction
to total flow. This regional setting is an important outcome to understand the streamflow variability in Peninsular
region of India.





*Data availability.* The streamflow datasets used for the analysis are accessible from
https://indiawris.gov.in/wris/#/. The daily India Meteorological Department (IMD) gridded rainfall product at
spatial resolution of 0.25° × 0.25°
(https://www.imdpune.gov.in/Clim_Pred_LRF_New/Grided_Data_Download.html) from Pai et al., (2014) is
used. The function baseflow, used for partitioning total flow to slow flow is downloaded from
https://in.mathworks.com/matlabcentral/fileexchange/58525-baseflow-filter-using-the-recursive-digital-filter-
technique.
*Author contributions.* PD, JM, and MS conceptualized the work, developed the methodology, and carried out the
data curation, formal analysis, validation, and writing of the original draft. MS and PPM reviewed the initial
manuscript, and PPM provided the resources needed for this work.
*Competing interests.* The authors declare that they have no conflict of interest.
*Acknowledgements.* PD acknowledges DST INSPIRE Faculty Fellowship (DST/INSPIRE/04/2022/001952
Faculty Reference No.: IFA22-EAS 114) received from Department of Science and Technology, Government of
India, in Earth and Atmospheric Sciences Division of 2022 call. MS acknowledges the award of a Satish Dhawan
Endowed Visiting Professorship that enabled him to visit the Interdisciplinary Centre for Water Research
(ICWaR) at the Indian Institute of Science, which allowed him to participate in the research activity that
culminated in this paper. MS also acknowledges the generous support and facilities provided by ICWaR that made
his stay a very productive one.

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
