# Peer review of "On the regional-scale variability of flow duration curves in"

_Hydrology and Earth System Sciences, 2023_

## Author Response (AR1)

| | |
|---|---|
| **Old Title** | : On the regional-scale streamflow variability using flow duration curve |
| **New Title** | : On the regional-scale variability of flow duration curves in Peninsular India |
| **MS No.** | : hess-2023-178 [Hydrology and Earth System Sciences] |
| **Authors** | : Pankaj Dey, Jeenu Mathai, Murugesu Sivapalan and Pradeep. P. Mujumdar |

We extend our sincere gratitude to the Editor, two anonymous reviewers, and Chris Leong for their thoughtful consideration of our manuscript and for providing valuable comments to enhance its quality. In response to the reviewers' feedback, we have diligently revised the manuscript, incorporating their suggestions to improve the introduction, study area, methodology, and conclusion sections. These modifications were made with the aim of effectively highlighting the novelty of our work. To maintain transparency, we have documented our responses to the reviewers' comments and outlined the corresponding changes in the revised manuscript. We believe that these enhancements have strengthened the overall coherence and impact of our contribution. We would like to express our appreciation to the Editor, anonymous reviewers, and Chris Leong for their invaluable input, which has undeniably enriched the final version of our manuscript. We sincerely thank them for their time and effort in reviewing our submission.

**RC1: 'Comment on hess-2023-178', Anonymous Referee #1, 30 Aug 2023**

The paper by Dey and coauthors "On the regional-scale streamflow variability using flow duration curve" explores the spatial patterns of flow duration curves across the Peninsular India, with the aim of identifying the main drivers of the water resources in a complex lowland region dominated by Monsoons. The analysis is performed by applying a "separation method" to disentangle fast and slow runoff components, and a "decomposition method" to express the annual FDC as a function of the underlying seasonal FDCs. The main results, as stated in the abstract and in the conclusions, pertain to the identification of the role of recession parameters and rainfall regime in the observed patterns of flow duration curves across the focus region.

While the study of FDCs in the Peninsular India is potentially interesting, the paper is very long (approximately 40 pages and 14 figures) and not always easy to read.

I must note that the quality of the Figures is not sufficient in most cases: some Figures contain a lot of panels with super-small labels and axis titles, which are nearly impossible to read and understand (e.g., Figure 2, 4, 5, 8 and 14).

From the perspective of the presentation, I think it would be necessary to implement the following key actions to improve the quality of this work and make it publishable:

Condense the MS text, focusing on the few real novel elements of this analysis

Improve the quality of the presentation, including clarity, typos, Figures quality, references – see below.

**Response:** We appreciate your valuable feedback on our manuscript and have diligently addressed your comments in the revised version. Our primary focus was on streamlining the content while ensuring the clarity of our contributions, figures, and overall narrative. In this comprehensive revision, we have carefully incorporated your suggestions to strengthen

sections such as the introduction, study area, methodology, and conclusion. The goal was to highlight the novelty of our work more effectively. We would like to extend our apologies for any inconvenience caused by the small labels and axis titles in Figures 2, 4, 5, 8, and 14. To address this issue, we have made improvements to Figure 2 and reformatted Figures 4, 5, 8, and 14 in 'landscape mode' to enhance readability. We believe that the revised manuscript now better reflects the novelty of our research in a concise and accessible manner.

The title of the paper is quite vague and does not tell nothing about the specific contribution of this paper. "Regional streamflow variability in Peninsular India" or "regional-scale variability of flow duration curves in Peninsular India" seem to be better options.

**Response:** Thank you for your valuable input, and we agree that a more precise title is essential. After careful consideration and thorough revision, we have changed the title to: "***On the regional-scale variability of flow duration curves in Peninsular India***".

In spite of the length of the Ms., the methods used are not new. The decomposition of timescales (Seasonal vs annual FDCs) – which is somewhat trivial from a statistical perspective - has been already applied in several past studies (e.g., Botter et al., WRR 2008; Muller et al., WRR 2014, Durighetto et al., WRR 2022). My impression is that this is not recognized in the text, and the approach is – implicitly - declared as a new contribution of the work (given the emphasis the authors put on this issue at the beginning of the methods).

**Response:** In our revised manuscript, we have now included the contributions of these studies, recognizing that the decomposition of timescales is not a novel concept in our study. While our approach aims to understand spatial patterns across Peninsular India, we acknowledge that the foundational idea is rooted in the previous works cited here. We have revised the text in the introduction section to provide a clearer connection to the existing literature and avoid any implicit declaration of novelty in the decomposition of timescales.

We acknowledge your concern regarding the lack of recognition for the prior application of the decomposition of timescales in several past studies, including Botter et al. (2008), Muller et al. (2014), and Durighetto et al. (2022).

Botter et al. (2008) addressed river basin streamflow variability by presenting a seasonal probability distribution for daily streamflow using a stochastic soil moisture model. Extending this to the annual scale, the study establishes analytical expressions for long-term flow duration curves, linking them to annual minima distribution through key basin parameters, including climate, ecohydrology, and geomorphology.

Muller et al. (2014) presents a process-based analytical expression for flow duration curves in seasonally dry climates, employing a stochastic model for wet season streamflow and a deterministic recession with stochastic initial conditions for the dry season. The approach disentangles inter- and intra-annual streamflow variations effectively.

Durighetto et al. (2022) develops analytical expressions for flow duration curves and stream length duration curves (SLDC) to classify streamflow and active length regimes in temporary rivers. It identifies two streamflow regimes (persistent and erratic) and three active length regimes (ephemeral, perennial, and ephemeral de facto) based on dimensionless parameters linked to streamflow fluctuations and catchment discharge sensitivity. The proposed framework, validated in Italy and USA catchments, reveals a structural relationship between streamflow and active length regimes, offering a promising tool for analysing discharge and river network length dynamics in temporary streams.

**References:**

- Botter, G., Zanardo, S., Porporato, A., Rodriguez-Iturbe, I., & Rinaldo, A. (2008). Ecohydrological model of flow duration curves and annual minima. *Water resources research*, *44*(8).
- Durighetto, N., Mariotto, V., Zanetti, F., McGuire, K. J., Mendicino, G., Senatore, A., & Botter, G. (2022). Probabilistic description of streamflow and active length regimes in rivers. Water Resources Research, 58, e2021WR031344. https://doi.org/10.1029/2021WR031344
- Müller, M. F., Dralle, D. N., & Thompson, S. E. (2014). Analytical model for flow duration curves in seasonally dry climates. *Water Resources Research*, *50*(7), 5510-5531.

Likewise, the decomposition of FDC into fast and slow components is the one proposed by Ghotbi et al., 2020 and 2021 (which in turn relates to concepts already well known in the literature, see e.g., Stewart, HESS 2015 and Leong and Yokoo, HYP 2022). Incidentally, this approach has its own limitations, provided that it is based on an empirical (subjective) filter to decompose the total streamflow into a fast and a slow component.

**Response:** While we initially employed the approach proposed by Ghotbi et al. (2020) and (2021) as a foundational method to characterize fast and slow flow components in our study, we acknowledge its inherent limitations due to its empirical and subjective nature. In the revised paper, we used recursive digital filter approach to decompose total flow into fast and slow flow components (in Supplementary Information). The selection of the filter parameter was done based on numerical experiments (in Supplementary Information). Recognizing these constraints, we incorporated, cited, and acknowledged the advancements of later-developed approaches, such as Stewart's (2015) Bump and Rise Method (BRM) and the approach by Leong and Yokoo (2022).

Stewart (2015) introduces the Bump and Rise Method (BRM), a novel baseflow separation technique calibrated with tracer data or optimization methods for accurate replication of tracer-determined baseflow shapes. The study challenges the conventional practice of solely relying on streamflow for recession analysis, contending that it can be misleading in understanding catchment storage reservoirs. The study also suggests for implementing baseflow separation before recession analysis as a means to gain fresh insights into water storage reservoirs and potentially resolve existing issues associated with recession analysis.

Significant advancements have been achieved in unravelling the process controls influencing flow duration curves. However, challenges persist in extending this knowledge to large spatial scales. To address this, Leong and Yokoo (2022) proposed an innovative approach, aiming to enhance the flexibility and adaptability of hydrological models by transforming the representation of the subsurface component. This involves the creation of a flexible structure composed of interconnected linear reservoirs, derived from a distinctive multiple hydrograph separation procedure, offering a comprehensive interpretation of dominant processes impacting FDC shapes and understand the number of distinct hydrological processes involved.

In the current revision, the controls of fast and slow flow components and their variation across space are highlighted.

**References:**

- Stewart, M. K. (2015). Promising new baseflow separation and recession analysis methods applied to streamflow at Glendhu Catchment, New Zealand. *Hydrology and Earth System Sciences*, *19*(6), 2587-2603.
- Leong, C., & Yokoo, Y. (2022). A multiple hydrograph separation technique for identifying hydrological model structures and an interpretation of dominant process controls on flow duration curves. *Hydrological Processes*, 36(4), e14569. https://doi.org/10.1002/hyp.14569

Thus, this paper represents a regional scale application of existing methods, with the reasonable aim of analyzing the spatial patterns of FDC in the Peninsular India. In the light of the scope of HESS (substantial new concepts, ideas, methods), Journal of Hydrology: regional study seems to be a better fit for this paper.

**Response:** We appreciate the concerns raised. The novelty of the paper lies in exploring the controls of streamflow variability in Peninsular India, a result of the impacts of monsoons – southwest (summer season) and northeast (winter season), the presence of western and eastern ghats, and topographical gradients. The paper advances the field by partitioning streamflow into three distinct time-wise categories (non-monsoon, southwest monsoon, and northeast monsoon) and two process-wise partitions (fast flow and slow flow), using flow duration curves as a tool. This approach allows for a detailed examination of the relative contributions of each season and process to the overall annual flow. The results, as presented in the study, illuminate the variability in contributions across selected river basins, shedding light on how these partitions evolve in each basin.

Furthermore, the integration of a comprehensive approach to analysing flow duration curves by incorporating a Mixed Gamma Distribution (MGD) to model both fast and slow flow components, along with seasonal and regional exploration, enhances the study's novelty, and the study uncovers the influence of climate, geology, and hydrological processes on MGD parameters, providing a nuanced understanding of flow duration curve shapes. The inclusion of links between MGD parameters and landscape properties, as well as the association between the midsection slope of the FDC and recession parameters, adds an additional layer of sophistication to the analysis. This provides a more robust understanding and offers insights into spatial variations, highlighting the integrated role of surface and subsurface processes in shaping the catchment's average flow regime

In summary, the study stands out for its innovative combination of time scale decomposition, process decomposition, and statistical analyses, offering a holistic exploration of the controls of streamflow variability in Peninsular India. The partitioning approach and the integration of statistical analysis contribute significantly to advancing our understanding of the complex interactions shaping streamflow patterns in this region.

If the revised manuscript doesn't align with the requirements, we are open to transferring it to other journals.

In terms of methods (and presentation of the methods), I have the following additional major concerns:

The first 16 equations of the paper are really trivial, and definitely not new. This needs to be recognized in the paper (and this part of the paper needs to be shortened).

**Response:** We appreciate your suggestion. We have implemented the recommended changes by incorporating relevant expressions (2, 3, 4, 5, 13, 14, 15, 16, 17) into the main text.

Simultaneously, we have moved supporting equations (6, 7, 8, 9, 10, 11, 12) to the supplementary material.

I really miss the very reason for which the decomposition of timescales is presented in the main text, while the decomposition into fast and slow runoff components is reported in the appendix. Both these methods are not new, and this paper presents an application of existing techniques. Both the key methods used in the Ms. (timescales separation and flow component separation) need to be summarized in the main body, although properly acknowledging previous works where these methods have been proposed /used.

**Response:** We value your recommendation. The Peninsular India has a unique climatological feature – occurrence of two monsoons – southwest and northeast monsoons, which brings moisture in this region. The timing and amount of rainfall received during these monsoons are different, along with high spatial variability across Peninsular India. The decomposition of time scales is performed to investigate how these monsoons affect the streamflow variability. Further, we have incorporated your feedback by relocating the text on decomposition into fast and slow flow components from the supplementary material to the main manuscript. We have now briefly summarized the time scale decomposition and process decomposition process in the main manuscript.

While we acknowledge that both methods are not novel, we have explicitly addressed this in the introduction section, emphasizing the paper's focus on the application of existing techniques. Furthermore, we have taken steps to ensure proper acknowledgment of previous works related to timescale separation and fast and slow flow separation, in the main manuscript. Please refer to the revised sections on introduction and methodology in the revised manuscript.

The recession analysis used in this paper is very basic, and most of the literature on the topic (which is huge) is overlooked. Recession analysis is a very delicate and tricky issue and has limits, artifacts and caveats that need to be recognized. The authors, therefore, need to better put this work in the context of the existing literature on recession analysis, including the following papers: Stewart, HESS 2015; Jachens at al., HESS 2020; Biswal and Marani, GRL 2011; Rupp and Selker, AWR 2006; Dralle et al., GRL 2015, Basso et al., AWR 2015)

**Response:** Thank you for providing the relevant papers in connection to the recession analysis. In the current paper, we used recursive digital filter approach for recession analysis where we have discussed the role of filter parameter in estimating fast and slow flow components (in Supplementary Information). The selection of the filter parameter was done based on numerical experiments (in Supplementary Information). We will incorporate the discussion of the recession analysis in the context of the relevant papers mentioned.

We appreciate the valuable insights provided by the suggested literature, including works by Stewart (2015), Jachens et al. (2020), Biswal and Marani (2010), Rupp and Selker (2006), Dralle et al. (2015), and Basso et al. (2015). Stewart's (2015) work introduces a baseflow separation method (BRM) that challenges the conventional understanding of recession analysis by emphasizing the need to account for the varying mixture of components, such as quick flow and baseflow, in streamflow. Jachens et al. (2020) underlines the limitations of traditional recession analysis, arguing that watershed properties should be evaluated based on individual recession events rather than collective recessions to avoid bias introduced by event inter-arrival time, magnitudes, and antecedent conditions. Moreover, Biswal and Marani (2010) establish a link between river network morphology and recession curve properties, highlighting the role of low-flow discharges in shaping channel networks. Rupp and Selker's (2006) proposed

method addresses artifacts in the plotting of recession data, advocating for proper scaling of time increments to accurately represent the relationship between the time rate of change in discharge and discharge itself. Dralle et al. (2015) identify and address a mathematical artifact in the empirical determination of recession parameters, providing a rescaling method to enhance the information content of fitted power laws. Lastly, Basso et al. (2015) emphasizes the importance of considering the non-linearity of the storage-discharge relation and studying recession patterns for individual events to explain heavy-tailed streamflow distributions.


More in general, the problem of identifying the main drivers of flow duration curves is quite old in hydrology, and I would say it has been already resolved/ discussed in several previous studies, including (but not limited to) Leong et al., HYP 2022; Carlier et al., JoH 2018; Botter et al., PNAS 2013; Basso et al., AWR 2015; Ye et al., HESS 2012; Fenicia et al., HYP 2014. This paper should summarize what is already known from the literature and indicate to what extent this paper advances the state of the art, providing a stronger link with other theoretical, modeling and regional scale studies.

**Response:** Thank you for providing valuable suggestions and comments. We have now incorporated the major insights gained from the existing studies pertaining to the identification of main drivers of flow duration curves in the manuscript.

[revised manuscript text omitted]

In line with the above comments, in terms of results / conclusions, the paper provides several arguments that are well known or somewhat tautological. For instance, the core part of the abstract states that: "Findings of the study indicate that [...] higher fraction of moderate to good groundwater potential zones explains the higher contribution of slow flow to total flow across north-south gradient of the region. Shapes of fast and slow FDCs are controlled by recession parameters."

I think everyone can agree on the above statements. The idea that good groundwater potential enhances slow flows is not particularly groundbreaking and should not be reported in the abstract.

**Response:** We appreciate you bringing up this matter. The mentioned line has been taken out from the abstract, and we have rephrased the abstract.

Likewise, the key role of recession parameters in the definition of the shape of the flow duration curve has been discussed by several previous studies (Mandal and Cunanne, INHS 2009; Botter et al., PNAS 2013; Muller et al., WRR 2014; Basso et al., AWR 2015; Arai et al., JoH 2021; Leong and Yokoo, HYP 2022; Leong and Yokoo, HRL 2019) which are not properly acknowledged here.

**Response:** We acknowledge the insightful contributions of Botter et al. (2013), Muller et al. (2014), Basso et al. (2015), Arai et al. (2021), Leong and Yokoo (2022; 2019) to the discourse on the pivotal role of recession parameters in shaping the flow duration curve. Botter et al. (2013) provides a comprehensive framework for characterizing hydrological regimes, emphasizing the influence of recession parameters in distinguishing erratic and persistent flow

patterns. Muller et al. (2014) contribute by quantitatively expressing catchment-scale attributes through recession parameters, highlighting their role in characterizing hydrological response times. Basso et al. (2015) emphasizes the non-linearity of storage-discharge relations and the impact of recession parameters on the emergence of heavy-tailed streamflow distributions. Arai et al. (2021) offer practical insights into predicting FDCs for run-of-river hydropower, utilizing recession parameters in analytical models. Additionally, Leong and Yokoo (2022; 2019) speculate on the minimal impact of catchment size on recession flow variability, paving the way for future investigations into the controls of catchment attributes. Leong et al.'s (2022) innovative approach to representing subsurface components demonstrates the ongoing efforts to enhance hydrological models, providing a deeper understanding of dominant processes influencing FDC shapes at large spatial scales. These studies collectively contribute to a better comprehension of the intricate relationship between recession parameters and FDC characteristics.


Were this work better placed in the context of the existing literature, and the presentation improved as per the above recommendations, the reader could better value the novel contribution of this paper, and the insight deriving from the regional-scale application presented in this HESSD manuscript. This would require extensive revisions and a deep restructuring of the Ms., but would ultimately lead to a shorter, more comprehensive, and stronger work, in which the specific insight provided by the authors and the links /connections with the existing literature are better articulated.

**Response:** We appreciate the constructive feedback and suggestions for improving the manuscript. We have extensively revised and restructured to better present the work within existing literature, improving the paper's clarity, and emphasizing the novel contributions.

In case the authors need more info on the references included in this review in a condensed format, they are encouraged to contact me through the Editors, and I will be happy to provide the full list in a standard format.

**Response:** We express gratitude for your willingness to assist. If the necessity arises in future revisions, we will certainly reach out for additional information on the references. Additionally, thank you for your patience in dedicating time to review our paper.

**RC2: 'Comment on hess-2023-178', Chris Leong, 03 Sep 2023**

This paper tests FDC partitioning into fast and slow flow FDCs to identify the process controls on catchments in the Indian peninsular.

Although it seems to claim to be a novel approach, the framework is not new. Perhaps the approach is new to the region, therefore, although I would recommend publication after attending to my suggestions below, I strongly suggest this paper be transferred to a regional journal.

My comments are listed below.

Title- Title is too general, needs improvement.

**Response:** We appreciate your insightful recommendation. The title has been updated to: "***On the regional-scale variability of flow duration curves in Peninsular India***".

Introduction

Line 56-72. It is good that the authors have cited the keystone FDC paper (Yokoo and Sivapalan, 2011) that disaggregates the FDC. However, there is repeated referencing in the paragraph which contributes to the long length of this paper. I suggest dedicating the paragraph to summarizing the keystone paper- so that it reduces citation repetition.

**Response:** Thank you for your input. We value your suggestion to condense the paragraph by providing a summary of the pivotal paper (Yokoo and Sivapalan, 2011) to eliminate redundant citations. We have now adjusted the paragraph accordingly. Please review the revised introduction section.

"Streamflow observed in rivers results from the complex interplay of various hydrological processes, including runoff generation, overland and subsurface flow, and evaporation. These processes operate across multiple time and space scales, responding to climatic inputs and interacting with heterogeneous landscape properties. Deciphering the controls on streamflow variability and understanding their manifestation in the FDC shape pose significant challenges (Cheng et al., 2012; Ghotbi et al., 2020b; Yokoo & Sivapalan, 2011).

To address this complexity, Yokoo and Sivapalan (2011) proposed a conceptual framework for unravelling the process controls of the FDC. They considered the Total Flow Duration Curve (TFDC) as a statistical summation of a Fast Flow Duration Curve (FFDC) and a Slow Flow Duration Curve (SFDC). The FFDC, representing a filtered version of precipitation variability, is influenced by rainfall intensity patterns and surface soil characteristics. In contrast, the SFDC reflects the competition between subsurface drainage and evapotranspiration, with seasonality and regional geology playing stronger roles (Yokoo & Sivapalan, 2011). This distinction between fast (surface runoff) and slow (subsurface streamflow and groundwater flow) flow time scales allows for a nuanced understanding of the process controls governing each component separately (Cheng et al., 2012; Yokoo & Sivapalan, 2011)."

Lines 60-62. "Therefore, there is a need for appropriate conceptual frameworks that can bring out these process controls of FDCs and generate deep insights into the governing principles underpinning observed variability."

This may need rephrasing, as I think it works the other way around---i.e., identify the process controls to deliver appropriate conceptual frameworks and generate deep insights into the governing principles underpinning the catchments variability.

**Response:** We appreciate your observation. We have now rephrased to: *"Therefore, identifying the process controls is essential to develop appropriate conceptual frameworks. This approach enables the generation of profound insights into the governing principles that underpin the observed variability in catchments."*

Lines 92-97 "The scientific novelty and methodological advancement of the paper lie in two interconnected aspects, which have not been adopted in the literature to date: (i) the timescale partitioning framework is used to study the relative contributions of different seasons to the FDC (repeated for fast and slow flow components), exploring how the relative contributions holistically vary across the whole region and using the framework to reconstruct the annual flow duration curve using seasonal flow duration curves, (ii) the Wegenerian approach in connecting the spatial variability of streamflow at a regional scale using flow duration curve"

In my opinion it is certainly not novel/new as similar studies have already been conducted especially (Chouaib et al., 2018, 2019 (https://doi.org/10.1016/j.jhydrol.2018.01.037, https://doi.org/10.1080/02626667.2019.1657233). To compare to other similar studies and identify what makes theirs novel or different from the others, the authors may want to look at Leong and Yokoo 2021 (https://doi.org/10.1016/j.jhydrol.2021.126984) that reviews the FDC from 2000-2020, as it is during this time that flow/time partitioning begun to generate more interest and development.

**Response:** The novelty of the paper lies in exploring the controls of streamflow variability in Peninsular India, a result of the impacts of monsoons – southwest (summer season) and northeast (winter season), the presence of western and eastern ghats, and topographical gradients. The paper advances the field by partitioning streamflow into three distinct time-wise categories (non-monsoon, southwest monsoon, and northeast monsoon) and two process-wise partitions (fast flow and slow flow), using flow duration curves as a tool. This approach allows for a detailed examination of the relative contributions of each season and process to the overall annual flow. The results, as presented in the study, illuminate the variability in contributions across selected river basins, shedding light on how these partitions evolve in each basin.

Furthermore, the integration of a comprehensive approach to analysing flow duration curves by incorporating a Mixed Gamma Distribution (MGD) to model both fast and slow flow components, along with seasonal and regional exploration, enhances the study's novelty, and the study uncovers the influence of climate, geology, and hydrological processes on MGD parameters, providing a nuanced understanding of flow duration curve shapes. The inclusion of links between MGD parameters and landscape properties, as well as the association between the midsection slope of the FDC and recession parameters, adds an additional layer of sophistication to the analysis. This provides a more robust understanding and offers insights into spatial variations, highlighting the integrated role of surface and subsurface processes in shaping the catchment's average flow regime

In summary, the study stands out for its innovative combination of time scale decomposition, process decomposition, and statistical analyses, offering a holistic exploration of the controls

of streamflow variability in Peninsular India. The partitioning approach and the integration of statistical analysis contribute significantly to advancing our understanding of the complex interactions shaping streamflow patterns in this region.

The studies conducted by Chouaib et al. (2018, 2019) focus on the regional variation of FDCs in the eastern United States, providing a process-based analysis of the interaction between climate and landscape properties. Chouaib et al. (2018) specifically advance the understanding of FDC regional variation by investigating how climate-landscape interactions influence the shape of FDCs. Their study emphasizes the role of soil infiltration rates in shaping FDCs, highlighting the impact of precipitation variability, catchment elevation, and soil properties on flow patterns. Chouaib et al. (2019) extend their exploration to the applicability of predicting daily FDCs in ungauged catchments using mean monthly runoff, considering hydroclimatic data from 73 catchments in the eastern USA. They identify constraints related to catchment flow variability and landscape properties when using mean monthly runoff to predict FDCs, emphasizing the importance of understanding the predominant runoff generation mechanism.

Leong and Yokoo (2021) present a comprehensive review of FDC literature from 2000 to 2020, aiming to enhance the global-scale applicability and transferability of FDC studies. They categorize existing studies into two groups: those developing prediction tools for specific regions and hydroclimates and those improving process understanding across larger spatio-temporal scales. The review identifies the dominance of empirical models in Group (1) studies, often limited to regions with well-established gauged networks. In contrast, Group (2) studies focus on process-based approaches to understand the controls on FDCs, aiming for broader applicability. Leong and Yokoo emphasize the need for a general framework for process-based modeling of FDCs and highlight the challenges and future research directions in achieving global-scale applicability in FDC studies.

Our study introduces a novel approach to understanding streamflow variability in Peninsular India by integrating time scale partitioning and process decomposition techniques, and statistical analyses. Unlike previous studies, we specifically focus on the unique regional influences, such as the interplay of monsoons, the presence of western and eastern ghats, and topographical gradients, which collectively shape streamflow patterns. This innovative methodology not only sets our work apart but also contributes valuable insights into the controls of streamflow variability in a region characterized by diverse climatic and topographic features. The partitioning approach and the integration of statistical analysis significantly contribute to advancing our understanding of the complex interactions shaping streamflow patterns in this region.

Nevertheless, I still think the only difference is the approach is novel (or advancement) to the study region (comparing to other FDC studies conducted in and around the surrounding

Himalayan region—could identify the novelty of the work in the area), but the idea and results are certainly not new. Therefore, I might suggest stating the novelty as such in addition to considering publishing as a regional paper. FDC literature is flooded with regionalization studies of similar nature, and the authors should not find difficulty identifying the strong points of their paper.

**Response:** Thank you for your thoughtful feedback. We appreciate your perspective on the novelty of our work. While we acknowledge that the general concept of studying flow duration curves and regionalization has been explored in various contexts, the uniqueness of our paper lies in the specific focus on Peninsular India. Our research delves into the intricacies of streamflow variability in this region, considering the interplay of monsoons, the geographical features of the western and eastern ghats, and topographical gradients. The application of time scale decomposition and process decomposition, and statistical analyses, coupled with the utilization of the FDC tool, adds a distinctive dimension to our study.

The novelty of the paper lies in exploring the controls of streamflow variability in Peninsular India, a result of the impacts of monsoons – southwest (summer season) and northeast (winter season), the presence of western and eastern ghats, and topographical gradients. The paper advances the field by partitioning streamflow into three distinct time-wise categories (non-monsoon, southwest monsoon, and northeast monsoon) and two process-wise partitions (fast flow and slow flow), using flow duration curves as a tool. This approach allows for a detailed examination of the relative contributions of each season and process to the overall annual flow. The results, as presented in the study, illuminate the variability in contributions across selected river basins, shedding light on how these partitions evolve in each basin.

Furthermore, the integration of a comprehensive approach to analysing flow duration curves by incorporating a Mixed Gamma Distribution (MGD) to model both fast and slow flow components, along with seasonal and regional exploration, enhances the study's novelty, and the study uncovers the influence of climate, geology, and hydrological processes on MGD parameters, providing a nuanced understanding of flow duration curve shapes. The inclusion of links between MGD parameters and landscape properties, as well as the association between the midsection slope of the FDC and recession parameters, adds an additional layer of sophistication to the analysis. This provides a more robust understanding and offers insights into spatial variations, highlighting the integrated role of surface and subsurface processes in shaping the catchment's average flow regime

The study stands out for its innovative combination of time scale decomposition, process decomposition, and statistical analyses, offering a holistic exploration of the controls of streamflow variability in Peninsular India. The partitioning approach and the integration of statistical analysis contribute significantly to advancing our understanding of the complex interactions shaping streamflow patterns in this region.

We recognize the abundance of literature in FDC studies, but we believe our contribution is valuable in providing a detailed understanding of the controls on streamflow variability in the context of Peninsular India.

Lines 82-106 is a long read; I suggest splitting the paragraphs.

**Response:** Thank you for the suggestion. We have divided the content into two paragraphs, as recommended.

Line 92-93. "The scientific novelty and methodological advancement of the paper lie in two interconnected aspects, which have not been adopted in the literature to date":

I disagree, it already has been adopted and applied in different regions. Rephrase??

**Response:** Thank you for bringing this to our attention. Following your correction, we have removed the mentioned sentence from the manuscript during the revision process.

Line 97-99. So, is the paper an extension of Ghobi et al, 2020a with time partitioning? If so, I don't see a problem statement in the MS, what is lacking in the Ghotbi paper that the authors are trying to address, did the study lack time inclusion? If not, then remove the citation for this part as it is unnecessary and confusing.

**Response:** Thank you for bringing up this concern. (Line 99) In response to your feedback, we have removed the specified line from the manuscript and made corresponding adjustments to the introduction section during the revision process. Please review the revised version of introduction section.

Method

Line 121-145 is long and needs splitting

**Response:** Thank you for your suggestion. In response, we have split the lengthy paragraph into two smaller ones. Please refer to the revised text in the study region.

Figure 1. Improve the visibility of the figure (i.e., font size of the legend and coordinates), and general map indicators (e.g., scale and direction). After going through all the figures, generally, improve the visibility of all figures. Also, is it appropriate to label (i.e., a, b, c, etc.) within the boundary of some of the figures---move them top-left of each sub-figure? There are too many figures in the MS- 14 figures, consider moving some to supplementary (e.g., Figure 6)

**Response:** We appreciate your valuable suggestions. In response to the overall number of figures, we have moved Figure 6 to the supplementary material. Apologies for any inconvenience caused by small labels and axis titles in Figures 2, 4, 5, 8, and 14. We have specifically improved the readability of Figure 2, and for Figures 4, 5, 8, and 14, we have optimized them in 'landscape mode' to enhance visibility.

Could section 2." Study region" -- be summarized and move some parts to supplementary material, as the main part of the method begins in Section 3. Otherwise, it just contributes to the long length of the paper.

**Response:** We appreciate your valuable suggestion. We have condensed Section 2, "Study Region," reducing the length of the text. Additionally, certain details have been relocated to the supplementary material to streamline the main body of the paper. Please review the revised manuscript.

In Section 3. The governing equations are not cited. Maybe I missed it but what are the demarcations of fast and slow flows or how did the authors decide what was fast and slow? This needs to mention because of the recession parameter analysis in Figure 12.

**Response:** Thank you for your valuable suggestion. In response to the reviewers' feedback, Section 3 has been extensively revised. We have implemented the recommended changes by incorporating relevant expressions (2, 3, 4, 5, 13, 14, 15, 16, 17) into the main text.

Simultaneously, we have moved supporting equations (6, 7, 8, 9, 10, 11, 12) to the supplementary material. Please refer to the updated text in Section 3.

Results

In the results section of the paper, I must argue-- are again not new, I struggle to find what stands out from the results, if the authors can clearly state some new findings would certainly strengthen the novelty claims. Perhaps figures 9 and 13 (and Lines 603-603) are proof that this paper should be published in a regional journal

The results section could be titled "results and discussions"??

**Response:** We appreciate your insightful feedback. Following your suggestions, we have updated the section title to 'Results and Discussions.' The text within the introduction, study area, methodology, and conclusions sections has been revised to clearly articulate the findings, addressing concerns about the novelty of the results. We believe that these revisions have enhanced the clarity and significance of the discussions provided in this particular section. Additionally, Figure 6 has been relocated to the Supplementary file.

Conclusion

Lines 600-602- "Motivated by this fact, in this study we outlined a framework and its suitability for understanding process controls of FDCs, which involved separating annual streamflow into seasonal flow components and constructing annual FDC using seasonal FDCs."

I could not clearly identify what was the framework was. It seems that the framework the authors are using has already been developed and just testing the efficacy in a different region (see Lines 602-603). I'm confused as to whether the paper uses an existing framework. or develops a new framework (if so, —could a paragraph in the results section explain/describe the new framework).

**Response:** Thank you for bringing this to our attention. The mentioned line (Lines 600-602) has been removed from the manuscript to eliminate confusion. We acknowledge and apologize for any misunderstanding caused.

The novelty of the paper lies in exploring the controls of streamflow variability in Peninsular India, a result of the impacts of monsoons – southwest (summer season) and northeast (winter season), the presence of western and eastern ghats, and topographical gradients. The paper advances the field by partitioning streamflow into three distinct time-wise categories (non-monsoon, southwest monsoon, and northeast monsoon) and two process-wise partitions (fast flow and slow flow), using flow duration curves as a tool. This approach allows for a detailed examination of the relative contributions of each season and process to the overall annual flow. The results, as presented in the study, illuminate the variability in contributions across selected river basins, shedding light on how these partitions evolve in each basin.

Furthermore, the integration of a comprehensive approach to analysing flow duration curves by incorporating a Mixed Gamma Distribution (MGD) to model both fast and slow flow components, along with seasonal and regional exploration, enhances the study's novelty, and the study uncovers the influence of climate, geology, and hydrological processes on MGD parameters, providing a nuanced understanding of flow duration curve shapes. The inclusion of links between MGD parameters and landscape properties, as well as the association between

the midsection slope of the FDC and recession parameters, adds an additional layer of sophistication to the analysis. This provides a more robust understanding and offers insights into spatial variations, highlighting the integrated role of surface and subsurface processes in shaping the catchment's average flow regime

In summary, the study stands out for its innovative combination of time scale decomposition, process decomposition, and statistical analyses, offering a holistic exploration of the controls of streamflow variability in Peninsular India. The partitioning approach and the integration of statistical analysis contribute significantly to advancing our understanding of the complex interactions shaping streamflow patterns in this region.

The conclusion needs to be summarized ---it is repetitive of the results section--just choose few outstanding points that need to be highlighted. And move some parts to introduction (e.g., Lines 664-678).

**Response:** We appreciate your valuable feedback. In response to your suggestions, we have condensed the conclusion section by highlighting key points from the results section. Additionally, we have removed lines 664-678 from the conclusion. Please refer to the revised conclusion for a more concise and focused summary of the study's main findings.

**RC3: 'Comment on hess-2023-178', Anonymous Referee #3, 07 Sep 2023**

General:

Flow duration curves (FDCs) are used widely in hydrological practice. They provide valuable information that can be used as catchment-specific signatures. The paper takes FDC as the topic to study by using a large set of streamflow data from 62 gauging stations extended over four neighbouring large river basins in southern half of India. The streamflow data set extending from 1965 to 2012 at daily time interval is accompanied with daily gridded rainfall data at fine spatial resolution. Conceptualization of the methodology is based on the similarity of FDC as a hydrological tool, and Cumulative Probability Distribution Function (CDF) as a statistical tool. The streamflow was divided into three timewise partitions (non-monsoon, SW monsoon and NE monsoon) and into two process wise partitions (fast flow and slow flow) to determine relative contribution of each season and process to the annual flow. Results show how the contribution of each timewise and process wise partitions varies over the river basins selected in the study. Based a multiple regression analysis, this variability was connected to the climatic and physical characteristics of river basin, and to the parameter sets of probability distribution functions fitted to the streamflow data. Summarized as above the study is worth to review.

**Response:** We appreciate the valuable feedback on the paper. Thank you for dedicating time to review our work; your insights have significantly contributed to the enhancement of our study.

Specific:

1. Introduction is a relatively concise and well written section with relevant literature summary. However, the novel (better we say the new) part of the study was not clearly expressed (Lines 86 onward where "the extension of the method" is mentioned + Line 97-99 where the main goal of the study is given and the extension of the process

partitioning (Ghotbi et al., 2020a) with the time partitioning was mentioned). A clearer way of presentation of the new/novel piece is required to avoid any misinterpretation about that the paper is a repetition of earlier studies.

**Response:** Thank you for this valuable suggestion. We have now incorporated the novelty of the study with a clear description in the introduction section. We have extensively revised the last section of the introduction to articulate the novelty more effectively. Please review the revised introduction section.

2. Equations (1) and (2) are linked to each other as given in the paper correctly. One detail is that: The cumulative probability in Equation (1) is the probability that the variable (Q) is less or equal to a given value of variable (q). The probability we get from FDC is the probability that the variable (Q) equals or exceeds a given value of variable (q). Then, to satisfy the most right side of Equation (2), the middle part of Equation must be P[Q>q] not P[Q>=q]. Then, this will not be in accordance with the definition of FDC. I am commenting about this not to say that Equations (1) and (2) are incorrect but to remind this detail, which is practically nil as we know that, for a continuous variable such as streamflow here, probability that the variable gets a given value is zero because of the infinite number of possible outcomes. It might be good to put this detail in a sentence after Equation (2).

**Response:** Thank you for your meticulous observation. We will include a clarifying sentence in the manuscript, addressing the point.

3. Reading the paper by shifting among the main text, appendix and supplement is not easy. My suggestion would be to put some pieces from the supplement to either directly into the text or give in the appendix by at the same removing well-known parts. Two examples; I would suggest moving Figure S3 into the main text to come in Line 276, which is needed to explain the estimation method. Also, I would suggest removing the well-known details of the baseflow separation and multiple regression in the Appendix (A1, A5). This comment is applied to the rest of Appendix and Supplementary.

**Response:** We appreciate your feedback on the paper's readability. To address this concern, extensive revisions have been made to the Introduction, Study Area, Methodology, Conclusions, Appendix, and Supplementary sections to ensure a smoother reading experience and improved clarity in conveying ideas. Furthermore, we have relocated well-known details of baseflow separation, MGD, etc from the Appendix to the supplementary material.

4. Section 4 needs a revision to better explain the use of methodology. In its current version, the section is not understandable probably because of shifting from time-scale partitioning to other types of partitioning, and because of skipping from the main text to the Appendix and to the Supplement.

**Response:** Recognizing the difficulty caused by the shifting between time-scale partitioning and other types of partitioning, as well as the transitions between the main text, Appendix, and Supplement, we have revised and restructured the methodology section. The changes aim to enhance clarity and coherence. Please refer to the revised manuscript for an improved presentation of the methodology.

5. While revising the paper, the authors may give less emphasis to the study area but more to express the generalization of the methodology, otherwise the paper will seem like a regional study.

   **Response:** Thank you for this valuable point. We have made every effort to reduce the study area description and instead emphasize the presentation of the employed methodology. Please review the modified sections - Study Area and Methods.

6. Figures' readability is low because of too many figures in one and small size of fonts. An improvement is needed for the readability.

   **Response:** We appreciate your feedback on the readability of the figures. Apologies for any inconvenience caused by small labels and axis titles in Figures 2, 4, 5, 8, and 14. To address this, we have enhanced the readability by improving the layout of Figure 2, as well as Figures 4, 5, 8, and 14, placing them in 'landscape mode.'

Corrections:

Line 165: Mistyped catchment area 2,60,000 km2

**Response:** Thank you for pointing out this correction. We will rectify this.

Line 169: Correct 'bout' as 'about'

**Response:** Thank you for bringing this correction to our attention. The word has now been rectified. Please refer Text T1 in the Supplementary Information.

Line 263: 'where, $m = 1, 2, \ldots, 12$, represents the index for months' not needed, m already defined in Line 257.

**Response:** Thank you for highlighting this correction. The repeated definition of the index for the months has been removed as per your suggestion.

Line 284: DELETE 'the'

**Response:** Thank you for this correction. We will now correct the line as suggested.

Line 293: to plot of the flows?

**Response:** Thank you for pointing this out. We will rectify this.

Line 520: Change '&' with 'and'

**Response:** Thank you for suggesting this improvement. We will do this replacement.

Line 671: (cf. Alfred Wegener, Sivapalan, 2018): not listed or not properly listed among the references.

**Response:** Thank you for pointing this out. In the updated conclusion, we have deleted the mentioned text along with its corresponding citation.

---

## Author Response (AR2)

**Responses to the reviewers' comments**

**New Title** : On the regional-scale variability of flow duration curves in Peninsular India
**MS No.** : hess-2023-178 [Hydrology and Earth System Sciences]
**Authors** : Pankaj Dey, Jeenu Mathai, Murugesu Sivapalan and Pradeep. P. Mujumdar

We deeply appreciate the Editor, along with the two anonymous reviewers and Dr. Chris Leong, for meticulously evaluating our manuscript and providing insightful feedback to enhance its quality. Their constructive comments have significantly contributed to refining the manuscript, resulting in an improved version. We have attentively addressed each reviewer's comments and incorporated the suggested revisions in the updated manuscript. The line numbers mentioned in this document are referred to the *track-change* version of the manuscript. We sincerely thank you all for your invaluable contributions.

**Anonymous Referee #1**

Comment: I think the paper has largely improved as compared to the original version. The authors have clearly done a huge effort to restructure the paper (thank you). I think the content is now presented in a fairer and clearer manner, and the paper could be accepted for publication in HESS. Scope limitations and value of the presented study are now better defined. I indicate some minor pending issues bellow.

My only suggestion for additional improvements concerns the presentation. I think the intro does not follow a clear logic path as the authors go back and forth between the state of the art and the research gap / novelty of the study. I suggest to modify the order of some statements to help the reader to follow the reasoning of the authors from the state of the art to the research novelty / question (e.g. shouldn't lines 189-201 be moved earlier?). Likewise, I think methods / results / discussion are not clearly separated (e.g. lines 465 -473 are not methods!). The quality of the Figures is sufficient but could be further improved. The abstract does not have a standard format (definition of the problem, relevance of the problem, methods and main results + perspectives) and is quite long.

Response: Thank you very much for your thoughtful feedback and kind words regarding the revised version of our paper. We deeply appreciate the recognition of the extensive effort invested in restructuring the manuscript. Your expertise and constructive criticism have been instrumental in refining the quality of our work, and we are truly grateful for your input.

In response to your insightful feedback, we have taken steps to enhance the clarity and coherence of our paper. Specifically, we have reorganized the content to ensure a smoother transition for the reader. This involved relocating the literature review section (Lines 188-213) to precede the explanation of the novel aspects of our paper (Lines 175-189), thereby facilitating a more seamless progression from background research to our study's unique contributions.

Moreover, we have revisited the citations and phrasing within the manuscript to better align with the flow of information. For instance, we have rephrased certain sentences to convey the intricate relationship more effectively between recession parameters and FDC characteristics, drawing on insights from studies by Botter et al. (2013), Muller et al. (2014), Basso et al. (2015), Arai et al. (2021), and Leong and Yokoo (2022; 2019). This refinement aims to underscore the crucial influence of recession parameters on hydrological systems and highlight our examination of the connection between MGD parameters and landscape properties through recession analysis.

Furthermore, we have updated the abstract of the paper to provide a clearer overview of its contents, emphasizing key elements such as the problem statement, relevance, methodology, significant findings, and their implications.

We express our heartfelt appreciation once more for your invaluable contributions to the enhancement of our paper. Your feedback has played a crucial role in shaping its final iteration.

**Referee #2: Dr. Chris Leong**

Comment: The abstract needs to have some key findings/results mentioned. It focusses too much on the explanation of the novelty.

Response: Following the recommendations provided, we have revised the abstract to incorporate the highlighting of the problem statement, relevance of the problems, methods, key results, and their implications, thereby achieving a more equitable portrayal of our research. Thank you once again for dedicating your time to reviewing our manuscript and offering constructive criticism.

**Anonymous Referee #3**

Comments: I can see that the authors have responded all my comments. I am satisfied with them all except for the revision in Introduction where the novel/new part of the paper is explained. The authors have moved from the literature to the novelty of the paper and then to the literature again (see three paragraphs in Lines 167-187 where the novelty is explained, and Lines 188-213 where literature is reviewed).

Furthermore, to me, the novelty of the paper is over-explained in these three paragraphs. It seems that the third paragraph (Lines 182-187) is repeating the first paragraph (Lines 167-173), and the second and third paragraphs (Lines 174-187) are more like discussion of the novelty.

As a solution, I would suggest the authors to (i) bring the literature together first (move Lines 188-213 up to come before Line 167), and (ii) explain the novelty of the paper (three paragraphs from Line 167 to Line 187) in one single concise paragraph. Upon acceptance of the paper, formatting would be needed.

Response: Thank you for bringing your concerns to our attention. We acknowledge the points you raised regarding the organization of the Introduction section. We have reorganized it by moving the literature review section (Lines 188-213) to precede the explanation of the novel aspects of our paper (Lines 167-187), ensuring a smoother transition from the literature review to the specific contributions of our study. Additionally, we have condensed the discussion of the novelty of our manuscript into a more concise single paragraph (lines: 175-189), aiming to better streamline the presentation of our key contributions.